# Collaboratively Learning Linear Models with Structured Missing Data

**Chen Cheng**[*]
Stanford University
chencheng@stanford.edu

**Gary Cheng**[*]
Stanford University
chenggar@stanford.edu

**John Duchi**
Stanford University
jduchi@stanford.edu

## Abstract

We study the problem of collaboratively learning least squares estimates for $m$ agents. Each agent observes a different subset of the features—e.g., containing data collected from sensors of varying resolution. Our goal is to determine how to coordinate the agents in order to produce the best estimator for each agent. We propose a distributed, semi-supervised algorithm COLLAB, consisting of three steps: local training, aggregation, and distribution. Our procedure does not require communicating the labeled data, making it communication efficient and useful in settings where the labeled data is inaccessible. Despite this handicap, our procedure is nearly asymptotically, local-minimax optimal—even among estimators allowed to communicate the labeled data such as imputation methods. We test our method on US Census data. We also discuss generalizations of our method to non-Gaussian feature settings, non-linear settings, and Federated Learning.

## 1 Introduction

Consider a set of agents that collect data to make predictions, where different agents may collect different features—because of different sensor availability or specialization—but wish to leverage shared structure to achieve better accuracy. Concretely, suppose we have $m$ agents, where each agent $i \in [m]$ observes $n$ samples of $(x_{i+}, y)$ where $x_{i+} \in \mathbb{R}^{d_i}$ is some subset of $x \in \mathbb{R}^d$. We set this as a regression problem where the data $(x, y)$ has the linear relationship $y = \langle x, \theta \rangle + \xi$ for some noise variable $\xi$. For example, these agents could be a network of satellites, each collecting data with a distinct set of sensors of varying resolution and specialization, with the purpose of estimating quantities like crop-yields [22], biomass [18], and solar-flare intensity [12]. Or these agents could be a group of seismic sensors, using acoustic modalities or accelerometers to predict whether an earthquake will occur [2]. Other examples may include networks of hospitals or phones [13]. In these settings, the agents can share information to collaboratively train a model; however, they are limited by communication bandwidth constraints, a situation satellites and seismic sensors often face due to radio frequency spectrum scarcity and interference [6, 20]. Without being too rigorous, we will define a communication efficient algorithm as one where the per-agent communication cost does not scale with $n$, $d$, or $m$. No dependence on the dataset size $n$ is suited for applications with significant data volume but limited communication resources. No dependence on the overall dimension $d$ of the data ensures that agents are not incentivized to turn away agents with access to a richer set of features that would improve statistical performance but increase $d$. Finally, no dependence on $m$ ensures that the algorithm is scalable to large collectives of agents. Can we construct a statistically optimal and communication efficient procedure to estimate $\theta$?

We answer in the affirmative and introduce our estimator COLLAB. COLLAB consists of three steps: local training on all agents, aggregation on a coordinating server, and distribution back to all agents. Our algorithm is communication-efficient: each agent $i \in [m]$ syncs twice with a coordinating server

---

[*]Equal contribution, authors ordered alphabetically by (last name, first name)

37th Conference on Neural Information Processing Systems (NeurIPS 2023).

and incurs communication cost scaling like $\Theta(d_i^2)$. We prove local minimax lower bounds which prove that COLLAB is (nearly) instance-optimal. We choose to study this problem in a stylized linear setting so that we can provide stronger guarantees for the algorithms we make. Indeed, our results which pair the exact asymptotic covariance of our estimator COLLAB with matching asymptotic local minimax lower bounds heavily rely on the linearity of our problem and would not be possible without strong structural assumptions. Having said this, the theory we develop for linear models does hint at potential methods for non-linear settings, which we discuss in Section 7. We also acknowledge privacy considerations are important for real world systems such as hospitals. We choose to focus instead on sensor settings where privacy is less of a concern. We leave adapting our results to privacy-sensitive settings to future work.

We compare our methods to single-imputation methods theoretically and empirically. We choose to baseline against imputation methods for three reasons. First, if we ignore communication constraints, our problem is a missing data problem, where formally the data is "missing at random" (MAR) [16]. MAR problems are well studied, so we know that imputation methods work well theoretically and in practice [25, 15]. Second, because we have instance-optimal lower bounds, we know that imputation methods are also optimal for our problem. Finally, because imputation methods use more information than the method we propose, imputation will serve as a "oracle" baseline of sorts.

**Contributions.** We briefly summarize our contributions.

1. We design a communication-efficient, distributed learning algorithm COLLAB which performs a weighted de-biasing procedure on the ordinary least squares estimator of each agent's data.

2. We show COLLAB is asymptotically locally minimax optimal among estimators which have access to the ordinary least squares estimator of each agent's data. We also show that with some additional assumptions, COLLAB is also asymptotically locally minimax optimal among estimators that have access to *all* of the training data of all agents.

3. We propose and develop theory for various baseline methods based on imputation. We compare the statistical error and communication cost of COLLAB against these baseline methods both theoretically and empirically on real and synthetic data.

4. We discuss generalizations of COLLAB for non-Gaussian feature settings and non-linear settings. We highlight open problems and identify possible directions for future work.

## 1.1 Related Work

**Missing data.** If we ignore the communication and computational aspects of our problem, the problem we study reduces to one of estimation with missing data. There has been a lot of work on this topic; please see [16] for an overview. The data in our problem is missing at random (MAR)—the missing pattern does not depend on the value of the data and is known given agent $i$. There are many approaches to handling missing data such as weighting and model-based methods [24]. Most related to our work are methods on single imputation. Schafer and Schenker [25] shows imputation with conditional mean is nearly optimal with special corrections applied. More recently, Chandrasekher et al. [3] show that single imputation is minimax optimal in the high dimensional setting. Another closely related popular approach is multiple imputation [23, 1]. Previous work [27, 29] has shown that multiple imputation in low dimensional settings produces correct confidence intervals under a more general set of assumptions compared to single imputation settings. However, we choose to focus on single imputation methods for two reasons. First, we are interested in estimation error and not confidence intervals, and our lower bounds show that single imputation has optimal estimation error for our setting. Second, in our problem context, multiple imputation would require more rounds of communication and consequently higher communication cost. Other methods for missing data include weighting and model-based methods.

**Distributed learning.** Learning with communication constraints is a well studied practical problem. We provide a couple of examples. Suresh et al. [26] study how to perform mean estimation with communication constraints. Duchi et al. [8] develop communication-constrained minimax lower bounds. Distributed convex optimization methods like Hogwild [21] have also been well studied. However, the works mentioned all concern the no-missing-data regime. A more relevant subfield of distributed learning is federated learning. In federated learning, a central server coordinates a

collection of client devices to train a machine learning model. Training data is stored on client devices, and due to communication and privacy constraints, clients are not allowed to share their training data with each other or the central server [13]. In the no-missing-features regime, optimization algorithms for federated optimization are well studied. There is also more theoretical work, which focus on characterizing communication, statistical, and privacy tradeoffs, albeit for a more narrow set of problems such as mean and frequency estimation [4]. More related to the missing data regime we consider is cross-silo federated learning [13] or vertical federated learning [30]. In this paradigm, the datasets on client machines are not only partitioned by samples but also by features. Researchers have studied this problem in the context of trees [5], calculating covariance matrices [14], k-means clustering [28], support vector machines [31], and neural nets [17]. Most related to our work is Gascón et al. [9], Hardy et al. [11]; they study how to privately perform linear regression in a distributed manner. However, unlike our work, these works focus more on developing algorithms with privacy guarantees rather than statistical ones.

## 2    Mathematical model

We assume we have $m$ agents that observes a subset of the dimensions of the input data $x \in \mathbb{R}^d$. Each agent $i$ has a "view" permutation matrix $\Pi_i^\top := \begin{bmatrix} \Pi_{i+}^\top & \Pi_{i-}^\top \end{bmatrix} \in \mathbb{R}^{d \times d}$. $\Pi_{i+} \in \mathbb{R}^{d_i \times d}$ describes which feature dimensions the agent sees, and $\Pi_{i-} \in \mathbb{R}^{(d-d_i) \times d}$ describes the dimensions the agent does not see. For a feature vector $x \in \mathbb{R}^d$ and corresponding label $y \in \mathbb{R}$, the $i$-th agent observes $(x_{i+}, y)$ where $x_{i+} := \Pi_{i+} x \in \mathbb{R}^{d_i}$. Each agent has $n$ fresh, independent observations (independent across agents) denoted as a matrix $X_{i+} \in R^{n \times d_i}$ and vector $y_i \in \mathbb{R}^n$. We let $X_{i-} \in \mathbb{R}^{n \times (d-d_i)}$ denote the unobserved dimensions of the input data $x$ drawn for the $i$-th agent, and we let $X_i \in \mathbb{R}^{n \times d}$ denote the matrix of input data $x$ drawn for the $i$-th agent, including the dimensions of $x$ unobserved by the $i$-th agent. To simplify discussions in the following sections, for any vector $v \in \mathbb{R}^d$ we use the shorthand $v_{i+} = \Pi_{i+} v$ and $v_{i-} = \Pi_{i-} v$. Similarly for any matrix $A \in \mathbb{R}^{d \times d}$ we denote by

$$A_{i+} = \Pi_{i+} A \Pi_{i+}^\top, \qquad\qquad A_{i-} = \Pi_{i-} A \Pi_{i-}^\top,$$
$$A_{i\pm} = \Pi_{i+} A \Pi_{i-}^\top, \qquad\qquad A_{i\mp} = \Pi_{i-} A \Pi_{i+}^\top.$$

For a p.s.d. matrix $A$, we let $\|x\|_A = \langle x, Ax \rangle$.

We assume the data from the $m$ agents follow the same linear model. The feature vectors $x$ comprising the data matrices $X_1, \ldots, X_m$ are i.i.d. with zero mean and covariance $\Sigma \succ 0$. We will assume that each agent has knowledge of $\Sigma_{i+}$—e.g., they have a lot of unlabeled data to use to estimate this quantity. We also assume for each index tuple pair $(j, k)$ for $1 \leq j, k \leq d$, there exists some agent that observes dimensions $j$ and $k$. This ensures the coordinating server can construct the complete covariance matrix $\Sigma$ from individual $\Sigma_{i+}$. The labels generated follow the linear model

$$y_i = X_i \theta + \xi_i, \qquad \xi_i \overset{\text{iid}}{\sim} \mathsf{N}(0, \sigma^2 I_n).$$

Throughout this work we consider a fixed ground truth parameter $\theta$.

**Objectives.**    We are interested in proposing a method of using the data of the agents to form an estimate $\hat{\theta}$ which minimizes the full-feature prediction error on a fresh sample $x \in \mathbb{R}^d$

$$\mathbb{E}_x[(\langle x, \hat{\theta} \rangle - \langle x, \theta \rangle)^2] = \|\hat{\theta} - \theta\|_\Sigma^2. \tag{1}$$

We are also interested in forming an estimate $\hat{\theta}_i$ which minimizes the missing-feature prediction error of a fresh sample $x_{i+} \in \mathbb{R}^{d_i}$ for agent $i$—i.e., $x_{i+} = \Pi_{i+} x$ where $x \in \mathbb{R}^d$ is fresh. Define $T_i := \begin{bmatrix} I_{d_i} & \Sigma_{i+}^{-1} \Sigma_{i\pm} \end{bmatrix} \Pi_i$ and the Schur complement $\Gamma_{i-} := \Sigma \backslash \Sigma_{i+} := \Sigma_{i-} - \Sigma_{i\mp} \Sigma_{i+}^{-1} \Sigma_{i\pm}$. The local test error is then

$$\mathbb{E}_x[(\langle x_{i+}, \hat{\theta}_i \rangle - \langle x, \theta \rangle)^2] = \|\hat{\theta}_i - T_i \theta\|_{\Sigma_{i+}}^2 + \|\theta_{i-}\|_{\Gamma_{i-}}^2 \tag{2}$$

We see that $\|\theta_{i-}\|_{\Gamma_{i-}}^2$ is irreducible error. The role of the operator $T_i$ is significant, as (2) shows that $T_i \theta$ is the best possible linear predictor for the $i$th agent[2]. Through this paper, we will also highlight the communication costs of the methods we consider. Recall that we would like our methods to have communication cost that do not scale with $n$ or $d$.

---

[2] Maybe surprisingly, $T_i \theta$ is better than naively selecting the subset of $\theta$ corresponding to the features observed by agent $i$—i.e., $\Pi_i \theta$. This is because $T_i \theta$ leverages the correlations between features.

# 3  Method

We begin with some intuition of why collaboration should help. Then we outline an approach of solving this problem for general feature distributions. The general approach is not immediately usable because it requires some knowledge of $\theta$, so we need to do some massaging. In Section 3.3, we show how to circumvent this issue in the Gaussian feature setting and introduce our method COLLAB. Adapting the general approach to other non-Gaussian settings is an open problem, but we discuss some potential approaches in Section 7.

## 3.1  Intuition

In order to build some intuition for how our approach leverages collaboration, we begin with some basic analysis of the no-collaboration solution: where each agent $i$ only performs ordinary least squares on their local data

$$\hat{\theta}_i = X_{i+}^{\dagger} y_i \overset{\text{(i)}}{=} (X_{i+}^{\top} X_{i+})^{-1} X_{i+}^{\top} y_i \overset{\text{(ii)}}{=} \theta_{i+} + \widehat{\Sigma}_{i+}^{-1} \widehat{\Sigma}_{i\pm} \theta_{i-} + (X_{i+}^{\top} X_{i+})^{-1} X_{i+}^{\top} \xi_i,$$

here $A^{\dagger}$ denotes the Moore–Penrose inverse for a general matrix $A$, and (i) holds whenever $\mathrm{rank}(X_{i+}) \geq d_i$. Because we focus on the large sample asymptotics regime ($n \gg d_i$), (i) will hold with probability $1$. Expanding out the $y_i$ gives equality (ii). Ignoring the zero-mean noise term, we see that $\theta$ must satisfy $\theta_{i+} + \widehat{\Sigma}_{i+}^{-1} \widehat{\Sigma}_{i\pm} \theta_{i-} - \hat{\theta}_i \approx 0$ for all $i \in [m]$. Our approach essentially tries to find such a $\theta$.

## 3.2  Our General approach

In particular, after each agent $i$ performing ordinary least squares on their local data to produce $\{\hat{\theta}_i\}_{i\in[m]}$, we recover an estimate of $\theta$ by performing a form of weighted empirical risk minimization parameterized by the positive definite matrices $W_i \in \mathbb{R}^{d_i \times d_i}$

$$\hat{\theta} = \hat{\theta}(W_1, \cdots, W_m) \coloneqq \operatorname*{argmin}_{\theta} \sum_{i=1}^{m} \left\| \theta_{i+} + \Sigma_{i+}^{-1} \Sigma_{i\pm} \theta_{i-} - \hat{\theta}_i \right\|_{W_i}^2. \tag{3}$$

We know by first order stationarity that $\hat{\theta} = \left( \sum_{i=1}^{m} T_i^{\top} W_i T_i \right)^{-1} \left( \sum_{i=1}^{m} T_i^{\top} W_i \hat{\theta}_i \right)$. We can show that $\hat{\theta}$ is a consistent estimate of $\theta$ regardless the choice of weighting matrices $W_i$. In fact, if the features $X_i, \xi_i$ are Gaussian, $\hat{\theta}$ is also unbiased. See Lemma B.1 in the Appendix for this result. While Lemma B.1 shows that the choice of weighting matrices $W_i$ does not affect consistency, the choice of weighting matrices $W_i$ does affect the asymptotic convergence rate of the estimator. In the next theorem, we show what the best performing choice of weighting matrices are. The proof is in Appendix B.2.

**Theorem 3.1.** *For any weighting matrices $W_i$, the aggregated estimator $\hat{\theta} = \hat{\theta}(W_1, \cdots, W_m)$ is asymptotically normal*

$$\sqrt{n} \left( \hat{\theta} - \theta \right) = \mathsf{N}(0, C(W_1, \cdots, W_m)),$$

*with some covariance matrix $C(W_1, \cdots, W_m)$. The optimal choice of weighting matrices is*

$$W_i^{\star} \coloneqq \Sigma_{i+} (\mathbb{E} \left[ x_{i+} \theta_{i-}^{\top} z_{i+} z_{i+}^{\top} \theta_{i-} x_{i+}^{\top} \right] + \sigma^2 \Sigma_{i+})^{-1} \Sigma_{i+},$$

*where $z_{i+} = x_{i-} - \Sigma_{i\mp} \Sigma_{i+}^{-1} x_{i+}$. In particular, for all $W_i$, $C(W_1, \cdots, W_m) \succeq C(W_1^{\star}, \cdots, W_m^{\star}) = \left( \sum_{i=1}^{m} T_i^{\top} W_i^{\star} T_i \right)^{-1}$.*

The main challenge of using Theorem 3.1 is in constructing the optimal weights $W_i^{\star}$, as at face value, they depend on knowledge of $\theta$. While we will discuss high level strategies of bypassing this issue in non-Gaussian data settings in Section 7, we will currently focus our attention on how we can make use of Gaussianity to construct our estimator COLLAB.

---

**Algorithm 1:** COLLAB algorithm

---

**Data:** $m$ agents with training data $(X_{1+}, y_1), \ldots, (X_{m+}, y_m)$ each with $n$ datapoints

**for** *Each agent $i = 1, \ldots, m$ in parallel* **do**

    Compute $\hat{\theta}_i = (X_{i+}^\top X_{i+})^{-1} X_{i+}^\top y_i$;

    Compute $\Sigma_i$ with (labeled and unlabeled) data;

    Compute $R_i = \frac{1}{n}\|X_{i+}\hat{\theta}_i - y\|_2^2$;

    Send $\hat{\theta}_i, \Sigma_i, R_i$ to coordinating server;

**end**

Coordinating server constructs $\Sigma$ from $\{\Sigma_j\}_{j=1}^m$;

Coordinating server constructs $\hat{W}_i^{\mathrm{g}} := \Sigma_{i+}/R_i$;

Coordinating server computes $\hat{\theta}_i^{\mathrm{clb}} = T_i\hat{\theta}(\hat{W}_1^{\mathrm{g}}, \cdots, \hat{W}_m^{\mathrm{g}})$ and distributes them to respective agents;

---

### 3.3 COLLAB Estimator - Gaussian feature setting

If $X_i$ are distributed as $\mathsf{N}(0, \Sigma)$, $W_i^\star$ has an explicit closed form as

$$W_i^\star = W_i^{\mathrm{g}} := \frac{\Sigma_{i+}}{\|\theta_{i-}\|_{\Gamma_{i-}}^2 + \sigma^2} = \frac{\Sigma_{i+}}{\mathbb{E}_{x,y}[(\langle x_{i+}, \hat{\theta}_i\rangle - y)^2]},$$

where $\Gamma_{i-} = \Sigma_{i-} - \Sigma_{i\mp}\Sigma_{i+}^{-1}\Sigma_{i\pm}$ is the Schur complement. Recall we assume that each agent has enough unlabeled data to estimate $\Sigma_{i+}$. The denominator $\mathbb{E}_{x,y}[(\langle x_{i+}, \hat{\theta}_i\rangle - y)^2]$ cannot be computed but we can use $\frac{1}{n}\|X_{i+}\hat{\theta}_i - y\|_2^2$, which is a consistent estimator of $\mathbb{E}_{x,y}[(\langle x_{i+}, \hat{\theta}_i\rangle - y)^2]$, in its place. Thus, each agent is able to construct estimates of $W_i^{\mathrm{g}}$ by computing

$$\hat{W}_i^{\mathrm{g}} := \frac{\Sigma_{i+}}{\frac{1}{n}\|X_{i+}\hat{\theta}_i - y\|_2^2}$$

Now we construct our global and local COLLAB estimators defined respectively as

$$\hat{\theta}^{\mathrm{clb}} := \hat{\theta}(\hat{W}_1^{\mathrm{g}}, \cdots, \hat{W}_m^{\mathrm{g}}), \qquad \hat{\theta}_i^{\mathrm{clb}} := T_i\hat{\theta}(\hat{W}_1^{\mathrm{g}}, \cdots, \hat{W}_m^{\mathrm{g}}). \tag{4}$$

We summarize the COLLAB algorithm in Algorithm 1. At a high level, $\hat{\theta}^{\mathrm{clb}}$ is an estimate of $\theta$ which also minimizes the full-feature prediction error (1) and $\hat{\theta}_i^{\mathrm{clb}}$ minimizes the missing-feature prediction error for agent $i$ (2). Now we show that using the collective "biased wisdom" of local estimates $\hat{\theta}_i$, our collaborative learning approach returns an improved local estimator. The proof is in Appendix B.3.

**Corollary 3.2.** *Let $X_i \sim \mathsf{N}(0, \Sigma)$ and define $C^{\mathrm{g}} := (\sum_{i=1}^m T_i^\top W_i^{\mathrm{g}} T_i)^{-1}$. The global COLLAB estimator $\hat{\theta}^{\mathrm{clb}}$ and the local $\hat{\theta}_i^{\mathrm{clb}}$ on agent $i$ are asymptotically normal*

$$\sqrt{n}\left(\hat{\theta}^{\mathrm{clb}} - \theta\right) \xrightarrow{d} \mathsf{N}\left(0, C^{\mathrm{g}}\right) \quad and \quad \sqrt{n}\left(\hat{\theta}_i^{\mathrm{clb}} - T_i\theta\right) \xrightarrow{d} \mathsf{N}\left(0, T_i C^{\mathrm{g}} T_i^\top\right).$$

*The following are true*

- *(i) $W_i^{\mathrm{g}}$ are the optimal choice of weighting matrices i.e.,particular, $C(W_1, \cdots, W_m) \succeq C(W_1^{\mathrm{g}}, \cdots, W_m^{\mathrm{g}}) = C^{\mathrm{g}}$.*

- *(ii) On agent $i$, we have $\sqrt{n}(\hat{\theta}_i - T_i\theta) \xrightarrow{d} \mathsf{N}(0, (W_i^{\mathrm{g}})^{-1})$. The asymptotic variance of $\hat{\theta}_i$ is larger than that of the COLLAB estimator $\hat{\theta}_i^{\mathrm{clb}}$—i.e., $(W_i^{\mathrm{g}})^{-1} \succeq T_i C^{\mathrm{g}} T_i^\top$.*

*Remark* 3.3. In Corollary 3.2, we can replace $\Sigma_{i+}$ in $\hat{W}_i^{\mathrm{g}}$ with local sample covariances $\hat{\Sigma}_{i+}$ for the plug-in estimator in Eq. (4). We prove this stronger statement in the proof of Corollary 3.2. However, as the coordinating server still need to use exact operators $T_i$ to compute the COLLAB estimator, relaxing this condition does not bring practical benefit.

| Method | Full-feature asymptotic covariance | Missing-feature asymptotic covariance | Communication cost for agent $i$ |
|---|---|---|---|
| Local OLS - $\hat{\theta}_i$ | - | $(W_i^{\mathrm{g}})^{-1}$ | $0$ |
| Local imputation w/ collaboration - $\hat{\theta}_i^{\mathrm{imp}}$ | $\left(\sum_{i=1}^m T_i^\top W_i^{\mathrm{g}} T_i\right)^{-1}$ | $T_i\left(\sum_{i=1}^m T_i^\top W_i^{\mathrm{g}} T_i\right)^{-1} T_i^\top$ | $d^2$ |
| Global imputation - $\hat{\theta}_i^{\mathrm{imp\text{-}glb}}$ | $\left(\sum_{i=1}^m T_i^\top W_i^{\mathrm{g}} T_i\right)^{-1}$ | $T_i\left(\sum_{i=1}^m T_i^\top W_i^{\mathrm{g}} T_i\right)^{-1} T_i^\top$ | $nd_i + d_i$ |
| COLLAB - $\hat{\theta}_i^{\mathrm{clb}}$ | $\left(\sum_{i=1}^m T_i^\top W_i^{\mathrm{g}} T_i\right)^{-1}$ | $T_i\left(\sum_{i=1}^m T_i^\top W_i^{\mathrm{g}} T_i\right)^{-1} T_i^\top$ | $d_i^2 + 2d_i + 1$ |

**Table 1.** Full and Missing feature asymptotic covariance and communication cost for agent $i$. Communication cost is measured by how many real numbers are received and sent from agent $i$.

This result characterizes the error distribution of the COLLAB estimator; guarantees about prediction error, defined in (1) and (2), directly follow. In particular, letting $z \in \mathbb{R}^d$ be distributed as $\mathsf{N}(0, C^{\mathrm{g}})$, by the continuous mapping theorem we have

$$n\mathbb{E}_x[(\langle x, \hat{\theta}^{\mathrm{clb}}\rangle - \langle x, \theta\rangle)^2] = n\|\hat{\theta}^{\mathrm{clb}} - \theta\|_\Sigma^2 \xrightarrow{d} \|z\|_\Sigma^2. \tag{5}$$

For large dimension $d$, $\|z\|_\Sigma^2$ concentrates around its expectation $\mathbb{E}[\|z\|_\Sigma^2] = \mathrm{Tr}(C^{\mathrm{g}})$. The same arguments can be applied to make a statement about the local test error (2).

**Communication Cost** In the first round of communication, agent $i$ sends $d_i^2 + d_i + 1$ real numbers to the coordinating server. In the second round of communication, the server sends back the updated local model which is $d_i$ real numbers. In total, this amounts to $d_i^2 + 2d_i + 1$ communication cost per agent. The communication cost of COLLAB does not scale with $n$, $d$, or $m$, satisfying our desiderata.

## 4 Comparison with other methods

In this section, we compare our collaborative learning procedure with other popular least squares techniques based on imputation and comment on the statistical efficacy and communication cost differences. We summarize our analysis in Table 1. The proofs of the theorems are in Appendix C. For the sake of brevity, our results here will center around estimating $\theta$. Analogous results centered around estimating $T_i\theta$ follow directly. These results can be connected to the full-feature and local test error by the same argument we made earlier to derive (5).

**Local imputation w/ collaboration.** Suppose a coordinating server collected covariance information $\Sigma_i$ from each agent and then distribute $\Sigma$ back to each of them. Then one intuitive strategy is to use this information to impute each agent's local data by replacing $X_{i+}$ with $\mathbb{E}[X_i \mid X_{i+}] = X_{i+}T_i$, before performing local linear regression. In other words, instead of computing $\hat{\theta}_i$, compute

$$\hat{\theta}_i^{\mathrm{imp}} = (T_i^\top X_{i+}^\top X_{i+} T_i)^\dagger T_i^\top X_{i+}^\top y_i$$

to send back to the coordinating server. Note that we use Moore–Penrose inverse here as $T_i^\top X_{i+}^\top X_{i+} T_i$ is in general of rank $d_i$, and $\hat{\theta}_i^{\mathrm{imp}}$ is then the min-norm interpolant of agent $i$'s data. Similar to COLLAB, we can use weighted empirical risk minimization parameterized by $W_i \in \mathbb{R}^{d \times d}$ and to aggregate $\hat{\theta}^{\mathrm{imp}}$ via

$$\hat{\theta}^{\mathrm{imp}} = \hat{\theta}(W_1, \cdots, W_m) \coloneqq \operatorname*{argmin}_\theta \sum_{i=1}^m \left\| T_i^\top (T_i T_i^\top)^{-1} T_i \theta - \hat{\theta}_i^{\mathrm{imp}} \right\|_{W_i}^2.$$

The next theorem, in conjunction with Theorem 3.1, implies that under the WERM optimization scheme, aggregation of least squares estimators on imputed local data does not bring additional statistical benefit. In fact, the local imputation estimator is a linearly transformed on local OLS $\hat{\theta}_i$.

**Theorem 4.1.** *For $\hat{\theta}_i^{\mathrm{imp}}$ from agent $i$, we have $\hat{\theta}_i^{\mathrm{imp}} = T_i^\top (T_i T_i^\top)^{-1} \hat{\theta}_i$. Given any weighting matrices $W_i \in \mathbb{R}^{d \times d}$, the aggregated imputation estimator $\hat{\theta}^{\mathrm{imp}}$ is consistent and asymptotically normal*

$$\sqrt{n}\left(\hat{\theta}^{\mathrm{imp}} - \theta\right) = \mathsf{N}(0, C^{\mathrm{imp}}(W_1, \cdots, W_m)).$$

*Using the same weights $W_i^\star \in \mathbb{R}^{d_i \times d_i}$ as in Theorem 3.1 for aggregated $\hat{\theta}^{\text{imp}}$, we have under p.s.d. cone order, for weights $W_i$, $C^{\text{imp}}(W_1, \cdots, W_m) \succeq C^\star$, where $C^\star = (\sum_{i=1}^m T_i^\top W_i^\star T_i)^{-1}$. In addition, the equality holds when $W_i = T_i^\top W_i^\star T_i$.*

As we will see in Sec. 5 where we provide minimax lower bound for weak observation models, the fact that the weighted imputation does not outperform our COLLAB approach is because the WERM on local OLS without imputation is already optimal. In fact, having access to the features will not achieve better estimation rate for both the global parameter $\theta$ and local parameters $T_i\theta$.

In terms of communication cost, this local imputation method requires more communication than COLLAB, as a coordinating server needs to communicate $\Sigma$ to all the agents. Each agent must first send $d_i^2$ real numbers corresponding to $\Sigma_i$ to the coordinating server, and then the coordinating server will send back $d^2 - d_i^2$ real numbers corresponding to the entries of $\Sigma$ that agent $i$ has not observed. This amounts to $d^2$ total communication cost per agent. The per-agent communication cost for this method scales with $d$, which is not desirable for the reason outlined in the introduction.

**Global imputation.** Finally, we analyze the setting where we allow each agent to send the coordinating server all of their data $(X_{i+}, y_i)$ for $i = 1, \cdots, m$ instead of their local estimators, $\hat{\theta}_i$ or $\hat{\theta}_i^{\text{imp}}$. Having all the data with structured missingness available, a natural idea is to have the coordinating server first impute the data, replacing $X_{i+}$ with $\mathbb{E}[X_i \mid X_{i+}] = X_{i+}T_i$, and then perform weighted OLS on *all* of the $nm$ data points, before sending a $d_i$ dimensional local model back to each agent. Namely for scalars $\alpha_1, \cdots, \alpha_m > 0$, we take

$$\hat{\theta}^{\text{imp-glb}} = \hat{\theta}^{\text{imp-glb}}(\alpha_1, \cdots, \alpha_m) := \left( \sum_{i=1}^m \alpha_i T_i^\top X_{i+}^\top X_{i+} T_i \right)^{-1} \left( \sum_{i=1}^m \alpha_i T_i^\top X_{i+}^\top y_i \right).$$

Surprisingly, in spite of the additional power, $\hat{\theta}^{\text{imp-glb}}$ still does not beat $\hat{\theta}$ in Theorem 3.1.

**Theorem 4.2.** *For any scalars $\alpha_1, \cdots, \alpha_m > 0$, $\hat{\theta}^{\text{imp-glb}}$ is consistent and asymptotically normal*

$$\sqrt{n} \left( \hat{\theta}^{\text{imp-glb}} - \theta \right) = \mathsf{N}(0, C^{\text{imp-glb}}(\alpha_1, \cdots, \alpha_m)).$$

*Recall the lower bound matrix $C^\star := (\sum_{i=1}^m T_i^\top W_i^\star T_i)^{-1}$ in Theorem 3.1. If $X_i \sim \mathsf{N}(0, \Sigma)$, we have under p.s.d. cone order and any $\alpha_i > 0$, $C^{\text{imp-glb}}(\alpha_1, \cdots, \alpha_m) \succeq C^\star$. In addition, the equality holds when $\alpha_i = 1/(\|\theta_{i-}\|_{\Gamma_{i-}}^2 + \sigma^2)$.*

The communication cost for this method is significantly larger than the other methods we discussed. Each agent must send all of its data to a coordinating server. Factoring in the additional communication cost of receiving a new local model from the server, this amounts to a total $nd_i + d_i$ communication cost per agent. The fact that communication cost for this method scales with $n$ is a significant disadvantage for the reason we outlined in the introduction.

## 5 Asymptotic Local Minimax Lower Bounds

In this section, we prove asymptotic local minimax lower bounds that show COLLAB is (nearly) optimal. We work in the partially-fixed-design regime. For every sample $x \in \mathbb{R}^d$, $x_{i+} \in \mathbb{R}^{d_i}$ is a fixed vector. We draw $x_{i-}$ from $\mathsf{N}(\mu_{i-}, \Gamma_{i-})$ where $\mu_{i-}$ and $\Gamma_{i-}$ is the conditional mean and variance of $x_{i-}$ given $x_{i+}$. Here $\Gamma_{i-}$ is also the Schur complement. We draw $x_{i-}$ from $\mathsf{N}(\mu_{i-}, \Gamma_{i-})$. The samples $x_{i+} \in \mathbb{R}^{d_i}$ comprise the matrices $X_{i+} \in \mathbb{R}^{n \times d_i}$. For all $i \in [m]$, we will assume we have an infinite sequence (w.r.t. $n$) of matrices $X_{i+}$. This partially-fixed-design scheme gives the estimators knowledge of the observed features and the distribution of the unobserved features, which is consistent with knowledge that COLLAB has access to. In this section we fix $\theta \in \mathbb{R}^d$. The corresponding label $y = x_{i+}\theta_{i+} + x_{i-}\theta_{i-} + \xi$, where $\xi \in \mathbb{R}$ is drawn from i.i.d. $\mathsf{N}(0, \sigma^2)$. We use $y_j \in \mathbb{R}^n$ to denote its vector form for the agent $j$. To model the estimator's knowledge about the labels, we will have two observation models—one weaker and one stronger—which we will specify later when we present our results.

For each observation model, we will have two types of results. The first type of result is a minimax lower bound for full-featured data; i.e., how well can estimator perform on a fresh sample without

missing features. This type of result will concern the full-feature asymptotic local minimax risk

$$\liminf_{n\to\infty} \mathfrak{M}_{m,\varepsilon}(\{X_{i+}\}_{i\in[m]}; \mathcal{P}_n, u) := \liminf_{n\to\infty} \inf_{\bar\theta} \sup_{P\in\mathcal{P}_n} n\mathbb{E}_{Z\sim P}\langle u, \bar\theta(Z, \{X_{i+}\}_{i\in[m]}) - \theta\rangle^2.$$

We will show that there exists a $B \in \mathbb{R}^{d\times d}$ such that the local minimax risk in the previous display is lower bounded by $u^T B u$ for all $u \in \mathbb{R}^d$. In other words, we have lower bounded the asymptotic covariance of our estimator with $B$ (with respect to the p.s.d. cone order). The second type of result is an agent specific minimax lower bound; i.e., what is the best prediction error an estimator (for the given observation model) can possibly have on a fresh sample for a given agent. This type of result will deal with the missing-feature asymptotic local minimax risk, defined as

$$\liminf_{n\to\infty} \mathfrak{M}_{m,\varepsilon}^{i+}(\{X_{i+}\}_{i\in[m]}; \mathcal{P}_n, u) := \liminf_{n\to\infty} \inf_{\bar\theta} \sup_{P\in\mathcal{P}_n} n\mathbb{E}_{Z\sim P}\langle u, \bar\theta(Z, \{X_{i+}\}_{i\in[m]}) - T_i\theta\rangle^2.$$

Similar to the first minimax error definition, we will again show that there exists a $B_i \in \mathbb{R}^{d_i\times d_i}$ such that the local minimax risk we just defined is lower bounded by $u^T B_i u$ for all $u \in \mathbb{R}^{d_i}$. Recall (2) for discussion surrounding why $T_i\theta$ is the right object to compare against.

## 5.1 Weak Observation Model: Access only to local models and features

Recall the local least squares estimator $\hat\theta_i = (X_{i+}^\top X_{i+})^{-1} X_{i+}^\top y_i$. Let $P_\theta^{\hat\theta}$ be a distribution over $\hat\theta_1, \ldots, \hat\theta_m$ induced by $\theta$ and $(\xi_1, \ldots, \xi_m) \overset{\text{iid}}{\sim} \mathsf{N}(0, \sigma^2 I_n)$. We define the following family of distributions $\mathcal{P}_{n,c}^{\hat\theta} := \{P_{\theta'}^{\hat\theta} : \|\theta' - \theta\|_2 \le cn^{-1/2}\}$ which defines our observation model. Intuitively, in this observation model, we are constructing a lower bound for estimators which have access to the features $X_{1+}, \ldots, X_{m+}$, the population covariance $\Sigma$, and access to $\hat\theta_1, \ldots, \hat\theta_m$. In comparison, our estimator COLLAB only uses $\Sigma$ and $\hat\theta_1, \ldots \hat\theta_m$. We present our first asymptotic local minimax lower bound result here. The proof of this result can be found in Appendix D.1.

**Theorem 5.1.** *Recall that $C^{\mathsf{g}} := (\sum_{i=1}^m T_i^\top W_i^{\mathsf{g}} T_i)^{-1}$. For all $\in [m]$ and $n$ let the rows of $X_{i+}$ be drawn i.i.d. from $\mathsf{N}(0, \Sigma_{i+})$. Then for all $u \in \mathbb{R}^d$, with probability 1, the full-feature asymptotic local minimax risk for $\mathcal{P}_{n,c}^{\hat\theta}$ is bounded below as,*

$$\liminf_{c\to\infty} \liminf_{n\to\infty} \mathfrak{M}_{m,\varepsilon}(\{X_{i+}\}_{i\in[m]}; \mathcal{P}_{n,c}^{\hat\theta}, u) \ge u^\top C^{\mathsf{g}} u.$$

*For all $u \in \mathbb{R}^{d_i}$, with probability 1, the missing-feature asymptotic local minimax risk for $\mathcal{P}_{n,c}^{\hat\theta}$ is bounded below as*

$$\liminf_{c\to\infty} \liminf_{n\to\infty} \mathfrak{M}_{m,\varepsilon}^{i+}(\{X_{i+}\}_{i\in[m]}; \mathcal{P}_{n,c}^{\hat\theta}, u) \ge u^\top T_i C^{\mathsf{g}} T_i^\top u.$$

This exactly matches the upper bound for COLLAB we presented in Corollary 3.2.

## 5.2 Strong Observation Model: Access to features and labels

Define the family of distributions $\mathcal{P}_{n,c}^y := \{P_{\theta'}^y : \|\theta' - \theta\|_2 \le cn^{-1/2}\}$ as the observation model. Intuitively, in this model, we are constructing a lower bound for estimators having access to all of the features $X_{1+}, \ldots, X_{m+}$ and access to $y_1, \ldots y_m$. This observation model is stronger than the previous observation model because estimators now have access to the labels $y$. We note again that our estimator COLLAB only uses $\Sigma$ and $\hat\theta_1, \ldots \hat\theta_m$. The quantities our estimator rely on do not scale with $n$, making our estimator much weaker than other potential estimators in this observation model, as estimators are allowed to depend on $y_i$, which grows in size with $n$. We present our second asymptotic local minimax lower bound result here, starting with defining the strong local lower bound matrix $C^{\mathsf{s}} := (\sum_{i=1}^m 2\Sigma/(\|\theta_{i-}\|_{\Gamma_{i-}}^2 + \sigma^2))^{-1}$. The proof of this result is in Appendix D.2.

**Theorem 5.2.** *For all $i \in [m]$ and $n$ let the rows of $X_{i+}$ be drawn i.i.d. from $\mathsf{N}(0, \Sigma_{i+})$. Then for all $u \in \mathbb{R}^d$, with probability 1, the full-feature asymptotic local minimax risk for $\mathcal{P}_{n,c}^y$ is bounded below as*

$$\liminf_{c\to\infty} \liminf_{n\to\infty} \mathfrak{M}_{m,\varepsilon}(\{X_{i+}\}_{i\in[m]}; \mathcal{P}_{n,c}^y, u) \ge u^\top C^{\mathsf{s}} u.$$

*For all $u \in \mathbb{R}^{d_i}$, with probability 1, the missing-feature asymptotic local minimax risk for $\mathcal{P}_{n,c}^y$ is bounded below as*

$$\liminf_{c\to\infty} \liminf_{n\to\infty} \mathfrak{M}_{m,\varepsilon}^{i+}(\{X_{i+}\}_{i\in[m]}; \mathcal{P}_{n,c}^y, u) \ge u^\top T_i C^{\mathsf{s}} T_i^\top u.$$

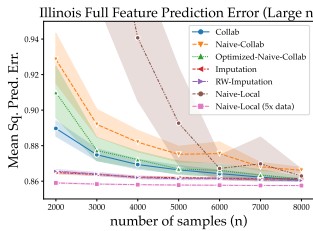 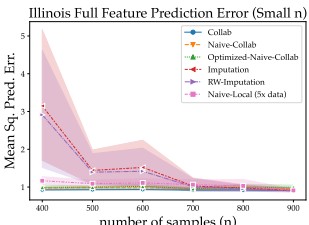 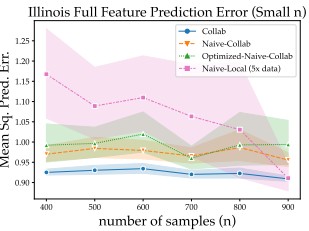

(a) Prediction Error for large $n$     (b) Prediction Error for small $n$     (c) Small $n$ omitting Impute algs.

**Figure 1:** Experimental results for US Census Experiment

In view of the lower bound in the strong observation model and that of the weak observation model in Theorem 5.1, it is clear that the lower bound in the strong observation setting is in general smaller as

$$\Sigma - T_i^\top \Sigma_{i+} T_i = \Pi_i^\top \begin{bmatrix} 0 & 0 \\ 0 & \Gamma_{i-} \end{bmatrix} \Pi_i \succeq 0,$$

which further implies $C^{\mathsf{g}} \succeq (\sum_{i=1}^m \Sigma/(\|\theta_{i-}\|_{\Gamma_{i-}}^2 + \sigma^2))^{-1} \succeq C^{\mathsf{s}}$.

We argue that the two lower bounds are comparable in the missing completely at random [16]. Consider for every agent $i$, each coordinate is missing independently with probability $p$. In this case, $(d_i, \Sigma_{i+}, T_i)$ are i.i.d. random triplets parameterized by $p$.

**Corollary 5.3.** *Under the random missingness setup with missing probability $p$, let the eigenvalue of $\Sigma$ be $\lambda_1(\Sigma) \geq \cdots \geq \lambda_d(\Sigma) > 0$ and define its condition number $\kappa = \lambda_1(\Sigma)/\lambda_d(\Sigma)$. Suppose $p \leq \frac{1}{2}\kappa^{-1}(1 + \|\theta\|_\Sigma^2/\sigma^2)^{-1}$, we have the limits $\lim_{m\to\infty} mC^{\mathsf{g}}$ and $\lim_{m\to\infty} mC^{\mathsf{s}}$ exist and*

$$4 \lim_{m\to\infty} mC^{\mathsf{s}} \succeq \lim_{m\to\infty} mC^{\mathsf{g}} \succeq \lim_{m\to\infty} mC^{\mathsf{s}}.$$

## 6 US Census Experiment

We experiment on real US census data modified from the ACSTravelTime dataset from the `folktables` package [7]. The code can be found at `https://github.com/garyxcheng/collab`. We (artificially) construct $m = 5$ datacenters (agents), each containing data from only one of California, New York, Texas, Florida, and Illinois. The goal is to collaboratively learn a model for each datacenter in a communication efficient way. This setup models potentially real settings where state governments are interested in similar prediction tasks but may not be allowed to directly transfer data about their constituents directly to one another due to privacy or communication constraints. In this setup, there is covariate shift across agents, and the feature and error distributions are not Gaussians—presenting significant deviations from our theoretical assumptions which serve as a proving ground to test how well COLLAB works in practice. For an experimental setup that more closely matches the assumptions we made in our theory, see Appendix A.2.

After dataset preprocessing, described in the Appendix A.1, we have $d = 37$ features. We compute the feature covariance matrix from training data across all the datacenters, which we plot in Appendix A.1. We assume we are able to do this because this computation can be done in a distributed manner, without communicating training data points or labels. The California datacenter will have access to 37 features, New York to 36, Texas to 35, Florida to 30, and Illinois to 27. This models the feature heterogeneity which varies across geography. Each datacenter will have $n$ datapoints, which we vary in this experiment. The objective is to predict people from Illinois's travel time to work given all 37 features. This task models the setting where the datacenter of interest does not have access to labeled full-featured, data to use to predict on full-featured test data.

We compare our method COLLAB against methods we call Naive-Local, Naive-Collab, Optimized-Naive-Colllab, Imputation, and RW-Imputation. We briefly describe each method here; Appendix A.1 contains a more detailed description of each method. Naive-Local refers to each agent locally perform OLS to construct $\hat{\theta}_i$. Naive-Collab does an equal-weighted average of the agent OLS models— $\sum_{i\in[m]} \Pi_{i+}^\top \hat{\theta}_i/m$. Optimized-Naive-Collab uses gradient descent to optimize the choices of weights used in Naive-Collab. Optimized-Naive-Collab uses fresh labeled samples without any missing

features during gradient descent, so in this sense, Optimized-Naive-Collab is more powerful than our method. Imputation refers to the global imputation estimator $\hat{\theta}^{\text{imp-glb}}$ with $\alpha_i = 1/m$. RW-Imputation is Imputation but with the optimal choice of weights $\alpha_i$. We also compare against Naive-Local trained with $5n$ datapoints. We choose $5n$ to model the hypothetical scenario setting where all of the other datacenters available contain data (albeit with missing features) from Illinois. For each method that we test, we run 80 trials to form 95% confidence intervals.

We see that for $n \leq 800$ in Figures 1(b) and 1(c), COLLAB performs the best; the imputation methods do the worst, and have much higher variance. In this small $n$ regime, even the Naive-Local method with 5 times the data does worse than COLLAB. For $n \geq 2000$ in Figure 1(a), the aggregation methods do worse than the imputation methods, and Naive-Local method with 5 times the data is the best performing method. However, COLLAB remains better than Optimized-Naive-Collab and Naive-Collab. The performance of Naive-Collab and Optimized-Naive-Collab being much closer to the performance of COLLAB here than in the synthetic data experiment in Appendix A.2 is not surprising, as the covariance of the features here is much more isotropic, meaning that the naive aggregation methods will not incur nearly as much bias.

## 7  Discussion and Future Work

**Optimal weights beyond Gaussianity.**  $\mathbb{E}\left[x_{i+}\theta_{i-}^\top z_{i+} z_{i+}^\top \theta_{i-} x_{i+}^\top\right]$ has a nice closed form in Gaussian setting because $z_{i+}$ and $x_{i+}$ are independent—which is in general not true without Gaussianity. If we can directly sample from the feature distribution $\mathcal{P}$ (e.g., unlabeled data), then we can empirically estimate $\mathbb{E}\left[x_{i+}\theta_{i-}^\top z_{i+} z_{i+}^\top \theta_{i-} x_{i+}^\top\right]$ by sampling from $\mathcal{P}$ and using any consistent plug-in estimate $\hat{\theta}$ (e.g., run COLLAB with weights $W_i = I_{d_i}$). This will return a good estimate of the optimal weights. An interesting future direction is to prove lower bounds without the Gaussianity assumption.

**Generalization to non-linear models.**  Recall in the Gaussian setting, the optimal weights in COLLAB are $W_i^{\text{g}} = \Sigma_{i+}/(\mathbb{E}_{x,y}[(\langle x_{i+}, \hat{\theta}_i \rangle - y)^2])$. Then, the optimal loss function in Eq. (3) becomes

$$\sum_{i=1}^m \left\| \theta_{i+} + \Sigma_{i+}^{-1}\Sigma_{i\pm}\theta_{i-} - \hat{\theta}_i \right\|_{W_i^{\text{g}}}^2 = \sum_{i=1}^m \frac{\mathbb{E}_{x_{i+}}[(\langle x_{i+}, \hat{\theta}_i \rangle - \langle x_{i+}, T_i\theta \rangle)^2]}{\mathbb{E}_{x,y}[(\langle x_{i+}, \hat{\theta}_i \rangle - y)^2]}.$$

This hints at a generalization to non-linear models. Suppose the local agents train on models $f^i(x_{i+}; \theta_i), \mathbb{R}^{d_i} \times \mathbb{R}^{d_i} \mapsto \mathcal{Y}$ and the global model $f(x; \theta), \mathbb{R}^d \times \mathbb{R}^d \mapsto \mathcal{Y}$ satisfies for some mapping $T_i : \mathbb{R}^d \to \mathbb{R}^{d_i}$, $f(x; T_i\theta) = f^i(x_{i+}; \theta_i)$. Consider a loss function $\ell(\cdot, \cdot) : \mathcal{Y} \times \mathcal{Y} \to [0, \infty)$. Then we can consider the following way of aggregation inspired by COLLAB for linear models

$$\hat{\theta} := \underset{\theta}{\arg\min} \sum_{i=1}^m \frac{\mathbb{E}_{x_{i+}}\ell(f^i(x_{i+}; \hat{\theta}_i), f(x_{i+}; T_i\theta))}{\mathbb{E}_{x_{i+}, y}\ell(f^i(x_{i+}; \hat{\theta}_i), y)}. \tag{6}$$

We can consistently estimate the denominators (weights) using training time loss. An interesting future direction is to investigate the performance of this general approach for non-linear problems.

**Application to Federated Learning.**  Federated Learning algorithms generally consist of repeating the following steps until termination: Step 1. server sends $\hat{\theta}$ to all agent, Step 2. each agent $i$ does some local training initialized at $\hat{\theta}$ to form $\hat{\theta}_i$, and Step 3. server collects $\hat{\theta}_i$ and aggregates them to form a new $\hat{\theta}$. Most federated learning algorithms assume the agent model sizes are the same across agents because aggregating models (Step 3) of different sizes is not obvious [19]. However, now (6) suggests a solution to this issue: in step 3, we can solve (6), albeit with population expectations replaced with empirical averages, to aggregate models of different sizes. Note that because data is assumed to stay with each agent in Federated Learning, this optimization problem may have to be solved via a federated learning algorithm.

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
