

**Figure 2:** Covariance Heatmap for US Census Experiment

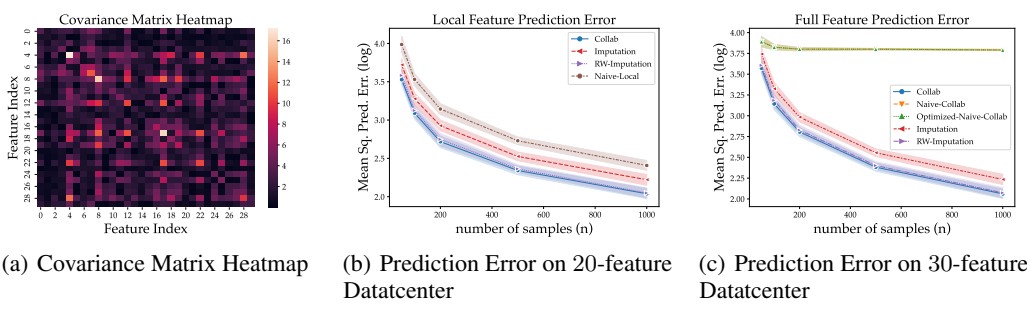

(a) Covariance Matrix Heatmap

(b) Prediction Error on 20-feature Datatcenter

(c) Prediction Error on 30-feature Datatcenter

**Figure 3:** Experimental results for Synthetic Experiment

## A Experimental Details

### A.1 Census Experimental Details

We use the 15 of the 17 features in the ACSTravelTime dataset—which include Age, Educational Attainment, Marital Status, Sex, Disability record, Mobility status, Relationship, etc. More specifically, using the notation from [7], we choose to keep the 'AGEP', 'SCHL', 'MAR', 'SEX', 'DIS', 'MIG', 'RELP', 'RAC1P', 'PUMA', 'CIT', 'OCCP', 'JWTR', 'POWPUMA', and 'POVPIP' features. We choose to exclude the State code (ST) and Employment Status of Parents (ESP) as a quick way to bypass low-rank covariance matrix issues. We turn the columns 'MAR', 'SEX', 'DIS', 'MIG', 'RAC1P', 'CIT', 'JWTR' into one-hot vectors. We make use commute time 'JWMNP' as the target variable. We clean our data by making sure AGEP (Age) must be greater than 16, PWGTP (Person weight) must be greater than or equal to 1, ESR (Employment status recode) must be equal to 1 (employed), and JWMNP (Travel time to work) is greater than 0. We normalize our features and targets by centering and dividing by the standard deviation computed from the training data. The California datacenter has access to all of the features. The New York datacenter has access to all categories except 'AGEP'. The Texas datacenter has access to all but 'AGEP', 'SCHL'. The Florida datacenter has access to all but 'AGEP', 'SCHL', 'MAR', 'SEX', and the Illinois datacenter has access to all but 'AGEP', 'SCHL', 'MAR', 'SEX', 'DIS', 'MIG'.

### A.2 Synthetic Experiments

We start with a synthetic experiment where we generate $m = 30$ agents observing some subset of $d = 30$ features. The code can be found at `https://github.com/garyxcheng/collab`. Ten of the agents will have access to random subsets of 20 of the features. The other twenty agents will have access to random subsets of 15 of the features. Each agent will have $n$ samples which we vary in this experiment. We sample the features from a $\mathsf{N}(0, \Sigma)$ distribution. We generate $\Sigma$ by first generating $d$ eigenvalues by sampling $d$ times from a uniform $[0, 1]$ distribution. We randomly select 3 eigenvalues to multiply by 10 and use these eigenvalues to populate the diagonal of a diagonal matrix $\Lambda$. Then we

use a randomly generated orthogonal matrix $W$ to form $\Sigma := W\Lambda W^T$. We plot a heatmap of $\Sigma$ in Figure 3(a). For each method that we test, we run 20 trials to form 95% confidence intervals.

We compare our method COLLAB, against the Imputation and RW-Imputation methods we outlined in Section 6. After we train each of these methods using the data on our 30 agents, we measure how well these methods perform in using the features of a test-agent with access to 20 of the total 30 features to predict outputs. We will also compare our methods against Naive-Local, where we only use the $n$ training datapoints of the 20 features our test-agent has access to, also described in Section 6. We plot this result in Figure 3(b).

We also compare our methods in an alternative setting where the test-center of interest has access to all 30 features. This setup models the setting where we are interested making the best possible predictions from all of the features available. In this experiment, we compare against Naive-Collab, Optimized-Naive-Collab, described in Section 6. We note that Optimized-Naive-Collab uses fresh labeled samples without any missing features during gradient descent, so in this sense, Optimized-Naive-Collab is more powerful than our method. We plot this result in Figure 3(c).

We see that reweighting is important; this is why COLLAB and RW-Imputation outperform the unweighted Imputation method. Our COLLAB method improves over the Naive-Local approach, meaning that the agents are benefiting from sharing information. COLLAB also matches the performance of the RW-Imputation method, despite only needing to communicate the learned parameters of each agent's model, as opposed to all of the data on each agent. The Naive-Collab approaches level out very quickly, likely reflecting the fact that these methods are biased, as the covariance of our underlying data is far from isotropic.

# B    Proofs for Section 3

**Lemma B.1.** *For any positive definite matrices $W_i \in \mathbb{R}^{d_i \times d_i}$, $i = 1, 2, \ldots, m$, the aggregated estimator $\hat{\theta}$ in Eq. (3) is consistent $\hat{\theta} \xrightarrow{P} \theta$. In addition, if $X_i \sim \mathsf{N}(0, \Sigma)$, we have unbiasedness $\mathbb{E}[\hat{\theta}] = \theta$ where $\mathbb{E}$ is over the random data $X_i$ and noise $\xi_i$.*

## B.1    Proof of Lemma B.1

For the general case, identify for $\hat{\theta}_i$, we can write

$$
\begin{aligned}
\hat{\theta}_i &= (X_{i+}^\top X_{i+})^{-1} X_{i+} y_i = (X_{i+}^\top X_{i+})^{-1} X_{i+}^\top (X_{i+}\theta_{i+} + X_{i-}\theta_{i-} + \xi_i) \\
&= \theta_{i+} + (X_{i+}^\top X_{i+})^{-1}(X_{i+}^\top X_{i-}\theta_{i-} + X_{i+}^\top \xi_i) \\
&= \theta_{i+} + \left(\frac{1}{n}X_{i+}^\top X_{i+}\right)^{-1}\left(\frac{1}{n}X_{i+}^\top X_{i-}\theta_{i-} + \frac{1}{n}X_{i+}^\top \xi_i\right).
\end{aligned}
$$

The weak law of large numbers implies that $X_{i+}^\top X_{i+}/n \xrightarrow{P} \Sigma_{i+}$, $X_{i+}^\top X_{i-}/n \xrightarrow{P} \Sigma_{i\pm}$ and $\frac{1}{n}X_{i+}^\top \xi_i \xrightarrow{P} 0$. Then Slutsky's theorem gives the consistency guarantee

$$
\hat{\theta}_i \xrightarrow{d} \theta_{i+} + \Sigma_{i+}^{-1}\left(\Sigma_{i\pm}\theta_{i-} + 0\right) = \theta_{i+} + \Sigma_{i+}^{-1}\Sigma_{i\pm}\theta_{i-} = T_i\theta,
$$

which is equivalent to $\hat{\theta}_i \xrightarrow{P} T_i\theta$. Substituting back into $\hat{\theta}$, we can obtain again from continuous mapping theorem that

$$
\hat{\theta} = \left(\sum_{i=1}^m T_i^\top W_i T_i\right)^{-1}\left(\sum_{i=1}^m T_i^\top W_i \hat{\theta}_i\right) \xrightarrow{p} \left(\sum_{i=1}^m T_i^\top W_i T_i\right)^{-1}\left(\sum_{i=1}^m T_i^\top W_i T_i \theta\right) = \theta.
$$

Next, we specialize to Gaussian features and show $\hat{\theta}$ is indeed unbiased in this case. By the tower property, we can write for each local OLS estimator,

$$
\begin{aligned}
\mathbb{E}[\hat{\theta}_i] &= \mathbb{E}[(X_{i+}^\top X_{i+})^{-1} X_{i+} y_i] = \mathbb{E}\left[\mathbb{E}[(X_{i+}^\top X_{i+})^{-1} X_{i+}^\top (X_{i+}\theta_{i+} + X_{i-}\theta_{i-} + \xi_i) \mid X_{i+}]\right] \\
&= \theta_{i+} + \mathbb{E}\left[(X_{i+}^\top X_{i+})^{-1} X_{i+}^\top \mathbb{E}[X_{i-} \mid X_{i+}]\right]\theta_{i-}.
\end{aligned}
$$

We want to compute $\mathbb{E}[X_{i-} \mid X_{i+}]$ and the key observation is that with Gaussianity in $X_i$, we have

$$
\mathsf{Cov}(x_{i-} - \Sigma_{i\mp}\Sigma_{i+}^{-1}x_{i+}, x_{i+}) = \mathsf{Cov}(x_{i-}, x_{i+}) - \Sigma_{i\mp}\Sigma_{i+}^{-1}\mathsf{Cov}(x_{i+}, x_{i+})
$$

$$= \Sigma_{i\mp} - \Sigma_{i\mp}\Sigma_{i+}^{-1} \cdot \Sigma_{i+} = 0,$$

and therefore $x_{i+}$ is independent of $x_{i-} - \Sigma_{i\mp}\Sigma_{i+}^{-1}x_{i+}$, which further implies that

$$\mathbb{E}\left[X_{i-} \mid X_{i+}\right] = \mathbb{E}\left[X_{i+}\Sigma_{i+}^{-1}\Sigma_{i\pm} \mid X_{i+}\right] + \mathbb{E}\left[X_{i-} - X_{i+}\Sigma_{i+}^{-1}\Sigma_{i\pm} \mid X_{i+}\right] = X_{i+}\Sigma_{i+}^{-1}\Sigma_{i\pm}.$$

Substituting the above property into computing the expectation of local estimates $\hat{\theta}_i$, it then holds

$$\mathbb{E}[\hat{\theta}_i] = \theta_{i+} + \mathbb{E}[(X_{i+}^\top X_{i+})^{-1}X_{i+}^\top X_{i+}\Sigma_{i+}^{-1}\Sigma_{i\pm}]\theta_{i-} = \theta_{i+} + \Sigma_{i+}^{-1}\Sigma_{i\pm}\theta_{i-} = T_i\theta.$$

We can then conclude the proof as

$$\mathbb{E}[\hat{\theta}] = \left(\sum_{i=1}^m T_i^\top W_i T_i\right)^{-1}\left(\sum_{i=1}^m T_i^\top W_i T_i\theta\right) = \theta.$$

## B.2 Proof of Theorem 3.1

We first study the central limit theorem for local OLS estimators $\hat{\theta}_i$. Let the data matrices $X_{i+} = [x_{i+}^1, \ldots, x_{i+}^n]^\top$ and $X_{i-} = [x_{i-}^1, \ldots, x_{i-}^n]$ and the noise vector $\xi_i = [\xi_i^1, \ldots, \xi_i^n]^\top$, we can write out for $\hat{\theta}_i$ that

$$\sqrt{n}\left(\hat{\theta}_i - T_i\theta\right) = \underbrace{(X_{i+}^\top X_{i+}/n)^{-1}}_{\text{(I)}} \cdot \underbrace{\frac{1}{\sqrt{n}}X_{i\pm}^\top\left\{(X_{i-} - X_{i+}\Sigma_{i+}^{-1}\Sigma_{i\pm})\theta_{i-} + \xi_i\right\}}_{\text{(II)}}. \tag{7}$$

For (II), note that

$$\frac{1}{\sqrt{n}}X_{i\pm}^\top\left\{(X_{i-} - X_{i+}\Sigma_{i+}^{-1}\Sigma_{i\pm})\theta_{i-} + \xi_i\right\} = \frac{1}{\sqrt{n}}\sum_{k=1}^n x_{i+}^j\left\{(x_{i-}^j - \Sigma_{i\mp}\Sigma_{i+}^{-1}x_{i+}^j)^\top\theta_{i-} + \xi_i^j\right\}.$$

The summands are independent mean zero random vectors, since

$$\mathbb{E}\left[x_{i+}^j\left\{(x_{i-}^j - \Sigma_{i\mp}\Sigma_{i+}^{-1}x_{i+}^j)^\top\theta_{i-}\right\}\right] = \left(\mathbb{E}\left[x_{i+}^j x_{i-}^{j\top}\right] - \mathbb{E}\left[x_{i+}^j x_{i+}^{j\top}\right]\Sigma_{i+}^{-1}\Sigma_{i\pm}\right)\theta_{i-}$$

$$= \left(\Sigma_{i\pm} - \Sigma_{i+}\Sigma_{i+}^{-1}\Sigma_{i\pm}\right)\theta_{i-} = 0,$$

and $\mathbb{E}[x_{i+}^j\xi_i^j] = \mathbb{E}[x_{i+}^j] \cdot \mathbb{E}[\xi_i^j] = 0$. Denote by $z_{i+}^j := x_{i-}^j - \Sigma_{i\mp}\Sigma_{i+}^{-1}x_{i+}^j$ and we can infer from the above display that $x_{i+}$ and $z_{i+}$ are uncorrelated. (II) is then asymptotically normal by CLT with limiting covariance (suppressing the superscript $j$ below)

$$\mathsf{Cov}\left(x_{i+}\left\{(x_{i-} - \Sigma_{i\mp}\Sigma_{i+}^{-1}x_{i+})^\top\theta_{i-} + \xi_i\right\}\right) = \mathbb{E}\left[x_{i+}\theta_{i-}^\top z_{i+}z_{i+}^\top\theta_{i-}x_{i+}^\top\right] + \mathbb{E}\left[\xi_i^2 x_{i+}x_{i+}^\top\right]$$

$$= \mathbb{E}\left[x_{i+}\theta_{i-}^\top z_{i+}z_{i+}^\top\theta_{i-}x_{i+}^\top\right] + \sigma^2\Sigma_{i+} := Q_i. \tag{8}$$

If $X_i$ are Gaussian random vectors, we can additionally have independence between $z_{i+}$ and $x_{i+}$ by zero correlation. Therefore

$$\mathbb{E}\left[x_{i+}\theta_{i-}^\top z_{i+}z_{i+}^\top\theta_{i-}x_{i+}^\top\right] = \mathbb{E}\left[x_{i+}\theta_{i-}^\top\mathbb{E}\left[z_{i+}z_{i+}^\top\right]\theta_{i-}x_{i+}^\top\right]$$

$$= \theta_{i-}^\top\mathsf{Cov}\left(x_{i-} - \Sigma_{i\mp}\Sigma_{i+}^{-1}x_{i+}\right)\theta_{i-} \cdot \mathbb{E}\left[x_{i+}x_{i+}^\top\right] = \theta_{i-}^\top\left(\Sigma_{i-} - \Sigma_{i\mp}\Sigma_{i+}^{-1}\Sigma_{i\pm}\right)\theta_{i-} \cdot \Sigma_{i+} = \|\theta_{i-}\|_{\Gamma_{i-}}^2\Sigma_{i+},$$

and $Q_i = (\|\theta_{i-}\|_{\Gamma_{i-}}^2 + \sigma^2)\Sigma_{i+}$.

We proceed to show $C(W_1, \cdots, W_n) \succeq C^\star$ under general feature distribution $\mathcal{P}$ and $W_i^\star := \Sigma_{i+}Q_i^{-1}\Sigma_{i+}$. By Slutsky theorem, (I) converges to $\Sigma_{i+}^{-1}$ in probability and we can conclude from Eq. (7) that

$$\sqrt{n}\left(\hat{\theta}_i - T_i\theta\right) \xrightarrow{d} \mathsf{N}\left(0, \Sigma_{i+}^{-1}Q_i\Sigma_{i+}^{-1}\right). \tag{9}$$

Further from $\hat{\theta} = \left(\sum_{i=1}^m T_i^\top W_i T_i\right)^{-1}\left(\sum_{i=1}^m T_i^\top W_i\hat{\theta}_i\right)$, it follows that

$$\sqrt{n}\left(\hat{\theta}_i - \theta\right) = \mathsf{N}(0, C(W_1, \cdots, W_n))$$

where

$$C(W_1, \cdots, W_n) = \left( \sum_{i=1}^{m} T_i^\top W_i T_i \right)^{-1} \cdot \left( \sum_{i=1}^{m} T_i^\top W_i W_i^{\star-1} W_i T_i \right) \cdot \left( \sum_{i=1}^{m} T_i^\top W_i T_i \right)^{-1}. \quad (10)$$

With the choice of $W_i = W_i^\star$, we achieve the claimed lower bound for asymptotic covariance as in this case $C(W_1, \cdots, W_m) = \left( \sum_{i=1}^{m} T_i^\top W_i^\star T_i \right)^{-1}$. It thus remains to show

$$C(W_1, \cdots, W_n) \succeq \left( \sum_{i=1}^{m} T_i^\top W_i^\star T_i \right)^{-1} = C^\star.$$

To prove the above claim, we construct auxiliary matrices $M_i$ as

$$M_i = \begin{bmatrix} T_i^\top W_i^\star T_i & T_i^\top W_i T_i \\ T_i^\top W_i T_i & T_i^\top W_i W_i^{\star-1} W_i T_i \end{bmatrix} = \begin{bmatrix} T_i^\top W_i^{\star\frac{1}{2}} \\ T_i^\top W_i W_i^{\star-\frac{1}{2}} \end{bmatrix} \begin{bmatrix} T_i^\top W_i^{\star\frac{1}{2}} \\ T_i^\top W_i W_i^{\star-\frac{1}{2}} \end{bmatrix}^\top \succeq 0.$$

Therefore

$$\sum_{i=1}^{m} M_i = \begin{bmatrix} C^{\star-1} & \sum_{i=1}^{m} T_i^\top W_i T_i \\ \sum_{i=1}^{m} T_i^\top W_i T_i & \sum_{i=1}^{m} T_i^\top W_i W_i^{\star-1} W_i T_i \end{bmatrix} \succeq 0.$$

As the Schur complement is also p.s.d. we can conclude with

$$0 \preceq C^{\star-1} - \left( \sum_{i=1}^{m} T_i^\top W_i T_i \right) \cdot \left( \sum_{i=1}^{m} T_i^\top W_i W_i^{\star-1} W_i T_i \right)^{-1}$$
$$\cdot \left( \sum_{i=1}^{m} T_i^\top W_i T_i \right) = C^{\star-1} - C(W_1, \cdots, W_n)^{-1}.$$

### B.3 Proof of Corollary 3.2

We prove the Corollary in the case of $\hat{W}_i^{\mathrm{g}} = \hat{\Sigma}_{i+}/\hat{R}_i$. When $\hat{W}_i^{\mathrm{g}} = \Sigma_{i+}/\hat{R}_i$ the proof is the same and slightly simpler. We first prove (i) and asymptotic normality of $\sqrt{n}(\hat{\theta}^{\mathrm{clb}} - \theta) \xrightarrow{d} \mathsf{N}(0, C^{\mathrm{g}})$. We point out that Theorem 3.1 is not directly applicable as we use estimated weights that reuse the training data. We claim consistency for $\hat{W}_i^{\mathrm{g}} \xrightarrow{p} W^{\mathrm{g}}$, and under this premise, the proof is rather straightforward since we can write

$$\sqrt{n}\left( \hat{\theta}^{\mathrm{clb}} - \theta \right) = \left( \sum_{i=1}^{m} T_i^\top \hat{W}_i^{\mathrm{g}} T_i \right)^{-1} \left( \sum_{i=1}^{m} T_i^\top \hat{W}_i^{\mathrm{g}} (\hat{\theta}_i - T_i \theta) \right).$$

With the asymptotic normality established for $\sqrt{n}(\hat{\theta}_i - T_i \theta)$ in Eq. (9), Slutsky's theorem and continuous mapping theorem, we can conclude that $\sqrt{n}(\hat{\theta}^{\mathrm{clb}} - \theta) \xrightarrow{d} \mathsf{N}(0, C^{\mathrm{g}})$. Now it remains to showing $\hat{W}_i^{\mathrm{g}} \xrightarrow{p} W^{\mathrm{g}}$, this is from Slutksy's theorem applied to $\hat{W}_i^{\mathrm{g}} = \hat{\Sigma}_{i+}/\hat{R}_i$ and the weak law of large numbers as follows

$$\hat{\Sigma}_{i+} = \frac{X_{i+}^\top X_{i+}}{n} \xrightarrow{p} \Sigma_{i+}, \qquad \hat{R}_i = \frac{1}{n} \|X_{i+}\hat{\theta}_i - y\|_2^2 \xrightarrow{p} \mathbb{E}[\|x_{i+}^\top T_i \theta - y_i\|_2^2],$$

where

$$\mathbb{E}[\|x_{i+}^\top T_i \theta - y_i\|_2^2] = \mathbb{E}[\|x_{i+}^\top \Sigma_{i+}^{-1} \Sigma_{i\pm} \theta_{i-} - x_{i-}^\top \theta_{i-}\|_2^2] + \sigma^2$$
$$= \|\theta_{i-}\|_{\mathsf{Cov}(x_{i-} - \Sigma_{i\mp} \Sigma_{i+}^{-1} x_{i+})}^2 + \sigma^2 = \|\theta_{i-}\|_{\Gamma_{i-}}^2 + \sigma^2.$$

We proceed to prove (ii). Applying delta method to the mapping $\theta \mapsto T_i \theta, \mathbb{R}^d \to \mathbb{R}^{d_i}$ on $\hat{\theta}(W_1^\star, \cdots, W_m^\star)$ immediately yields the asymptotic normality for $\hat{\theta}_i^{\mathrm{clb}}$. It only remains to show $T_i C^\star T_i^\top \preceq W_i^{\star-1}$.

Identify $W_i^{\star-1} - T_i C^\star T_i^\top$ as the Schur complement for the block matrix

$$M = \begin{bmatrix} W_i^{\star-1} & T_i \\ T_i^\top & C^{\star-1} \end{bmatrix},$$

and it suffices to show $M \succeq 0$. This follows from $C^\star = (\sum_{i=1}^m T_i^\top W_i^\star T_i)^{-1}$ and thus

$$M = \begin{bmatrix} W_i^{\star-1} & T_i \\ T_i^\top & \sum_{j=1}^m T_j^\top W_j^\star T_j \end{bmatrix} \succeq \begin{bmatrix} W_i^{\star-1} & T_i \\ T_i^\top & T_i^\top W_i^\star T_i \end{bmatrix} = \begin{bmatrix} W_i^{\star-\frac{1}{2}} \\ T_i^\top W_i^{\star\frac{1}{2}} \end{bmatrix} \begin{bmatrix} W_i^{\star-\frac{1}{2}} \\ T_i^\top W_i^{\star\frac{1}{2}} \end{bmatrix}^\top \succeq 0.$$

## C  Proofs for Section 4

### C.1  Proof of Theorem 4.1

The key part of the proof is showing $\hat{\theta}_i^{\text{imp}} = T_i^\top (T_i T_i^\top)^{-1} \hat{\theta}_i$. If we can have this claim established, we can make use of the following transformation of the loss function

$$\sum_{i=1}^m \left\| T_i^\top (T_i T_i^\top)^{-1} T_i \theta - \hat{\theta}_i^{\text{imp}} \right\|_{W_i}^2 = \sum_{i=1}^m \left\| T_i^\top (T_i T_i^\top)^{-1} T_i \theta - T_i^\top (T_i T_i^\top)^{-1} \hat{\theta}_i \right\|_{W_i}^2$$

$$= \sum_{i=1}^m \left\| T_i \theta - \hat{\theta}_i \right\|_{(T_i T_i^\top)^{-1} T_i W_i T_i^\top (T_i T_i^\top)^{-1}}^2 .$$

This reduces the optimization problem into the same one in Eq. (3) up to weight transformation, and the same lower bound for asymptotic covariance in Theorem 3.1 applies. Hence

$$C^{\text{imp-glb}}(\alpha_1, \cdots, \alpha_m) \succeq C^\star.$$

By taking $W_i = T_i^\top W_i^\star T_i$, we have the transformed weights satisfy

$$(T_i T_i^\top)^{-1} T_i W_i T_i^\top (T_i T_i^\top)^{-1} = (T_i T_i^\top)^{-1} T_i^\top W_i^\star T_i (T_i T_i^\top)^{-1} = W_i^\star.$$

From the optimality condition in Theorem 3.1, the equality holds under this choice of $W_i$'s.

It then boils down to proving the claim $\hat{\theta}_i^{\text{imp}} = T_i^\top (T_i T_i^\top)^{-1} \hat{\theta}_i$. We make use of the following two properties of Moore-Penrose pseudo inverse—for $A \in \mathbb{R}^{d_i \times d}$ of rank $d_i$,

$$(A^\top A)^\dagger = A^\dagger (A^\dagger)^\top, \qquad A^\dagger = A^\top (A A^\top)^{-1}.$$

Substituting $A = (X_{i+}^\top X_{i+})^{\frac{1}{2}} T_i$ into the above displays, we then have

$$\hat{\theta}_i^{\text{imp}} = (T_i^\top X_{i+}^\top X_{i+} T_i)^\dagger T_i^\top X_{i+}^\top y_i$$

$$= T_i^\top (X_{i+}^\top X_{i+})^{\frac{1}{2}} \left( (X_{i+}^\top X_{i+})^{\frac{1}{2}} T_i T_i^\top (X_{i+}^\top X_{i+})^{\frac{1}{2}} \right)^{-2} \cdot (X_{i+}^\top X_{i+})^{\frac{1}{2}} T_i T_i^\top X_{i+}^\top y_i$$

$$= T_i^\top (X_{i+}^\top X_{i+})^{\frac{1}{2}} \left( (X_{i+}^\top X_{i+})^{-\frac{1}{2}} (T_i T_i^\top)^{-1} (X_{i+}^\top X_{i+})^{-\frac{1}{2}} \right)^2 \cdot (X_{i+}^\top X_{i+})^{\frac{1}{2}} T_i T_i^\top X_{i+}^\top y_i$$

$$= T_i^\top (T_i T_i^\top)^{-1} \cdot (X_{i+}^\top X_{i+})^{-1} \cdot (T_i T_i^\top)^{-1} \cdot T_i T_i^\top X_{i+}^\top y_i$$

$$= T_i^\top (T_i T_i^\top)^{-1} \cdot (X_{i+}^\top X_{i+})^{-1} X_{i+}^\top y_i = T_i^\top (T_i T_i^\top)^{-1} \hat{\theta}_i.$$

### C.2  Proof of Theorem 4.2

By a direct calculation, we have

$$\hat{\theta}^{\text{imp-glb}} - \theta = \left( \sum_{i=1}^m \alpha_i T_i^\top X_{i+}^\top X_{i+} T_i \right)^{-1} \left( \sum_{i=1}^m \alpha_i T_i^\top X_{i+}^\top y_i \right) - \theta$$

$$= \left( \sum_{i=1}^m \alpha_i T_i^\top X_{i+}^\top X_{i+} T_i \right)^{-1} \left( \sum_{i=1}^m \alpha_i T_i^\top X_{i+}^\top (X_{i+} \theta_{i+} + X_{i-} \theta_{i-} + \xi_i) \right) - \theta$$

$$= \left( \sum_{i=1}^{m} \alpha_i T_i^\top X_{i+}^\top X_{i+} T_i \right)^{-1} \left( \sum_{i=1}^{m} \alpha_i T_i^\top X_{i+}^\top (X_{i+}\theta_{i+} + X_{i-}\theta_{i-} - X_{i+}T_i\theta + \xi_i) \right)$$

$$= \left( \sum_{i=1}^{m} \alpha_i T_i^\top X_{i+}^\top X_{i+} T_i \right)^{-1} \left( \sum_{i=1}^{m} \alpha_i T_i^\top X_{i+}^\top (X_{i-}\theta_{i-} - X_{i+}\Sigma_{i+}^{-1}\Sigma_{i\pm}\theta_{i-} + \xi_i) \right).$$

Consequently

$$\sqrt{n}\left( \hat{\theta}^{\text{imp-glb}} - \theta \right) = \left( \sum_{i=1}^{m} \alpha_i T_i^\top \cdot \frac{1}{n} X_{i+}^\top X_{i+} \cdot T_i \right)^{-1} \cdot \left( \sum_{i=1}^{m} \alpha_i T_i^\top \cdot \frac{1}{\sqrt{n}} X_{i+}^\top (X_{i-}\theta_{i-} - X_{i+}\Sigma_{i+}^{-1}\Sigma_{i\pm}\theta_{i-} + \xi_i) \right)$$

Following the same proof steps applied to Eq. (7) in Appendix B.2, we can conclude that

$$\sqrt{n}\left( \hat{\theta}^{\text{imp-glb}} - \theta \right)$$

$$\xrightarrow{d} \mathsf{N}\left( 0, \underbrace{\left( \sum_{i=1}^{m} \alpha_i T_i^\top \Sigma_{i+} T_i \right)^{-1} \left( \sum_{i=1}^{m} \alpha_i^2 T_i^\top Q_i T_i \right) \left( \sum_{i=1}^{m} \alpha_i T_i^\top \Sigma_{i+} T_i \right)^{-1}}_{:= C^{\text{imp-glb}}(\alpha_1, \cdots, \alpha_m)} \right),$$

with the same $Q_i$'s as in Eq. (8), and with Gaussianity of $X_i$, we also have the explicit form $Q_i = (\|\theta_{i-}\|_{\Gamma_{i-}}^2 + \sigma^2)\Sigma_{i+}$. Note that if $\alpha_i = 1/(\|\theta_{i-}\|_{\Gamma_{i-}}^2 + \sigma^2)$,

$$C^{\text{imp-glb}}(\alpha_1, \cdots, \alpha_m) = \left( \sum_{i=1}^{m} \frac{T_i^\top \Sigma_{i+} T_i}{\|\theta_{i-}\|_{\Gamma_{i-}}^2 + \sigma^2} \right)^{-1} = C^{\text{g}} = C^\star.$$

Finally, to show $C^{\text{imp-glb}}(\alpha_1, \cdots, \alpha_m) \succeq C^\star$, we identify from Eq. (10) that

$$C^{\text{imp-glb}}(\alpha_1, \cdots, \alpha_m) = C(\alpha_1 \Sigma_{1+}, \cdots, \alpha_m \Sigma_{m+}) \succeq C^\star,$$

where the last inequality follows from Theorem 3.1.

# D   Proofs for Section 5

We will use the van Trees inequality to prove our lower bound shown. In particular, we will use a slight modification to Theorem 4 of [10], which we state as a corollary below here. Throughout this section, we let $\psi : \mathbb{R}^d \to \mathbb{R}^s$ be an absolutely continuous function. The distribution $P_\theta$ in the family $\{P_\theta\}_{\theta \in \mathbb{R}^d}$ is assumed to have density $p_\theta$ which satisfies $\int_{\mathbb{R}^d} \|\nabla p_\theta(x)\|_2^2 \, dx < \infty$. Let $P_\theta^j$ for $j \in [m]$ denote the distribution over either $\tilde{\theta}_j^n$ or $y_j \in \mathbb{R}^n$. Let $\mathcal{I}_i^n(\theta)$ denote the Fisher Information of $P_\theta^i$, and let $\mathcal{I}^n(\theta) = \sum_{i=1}^{m} \mathcal{I}_i^n(\theta)$ denote the Fisher Information of $P_\theta$. We note that $P_\theta$ is allowed to depend on $n$.

**Corollary D.1** (Gassiat [10]). *Let $\psi : \mathbb{R}^d \to \mathbb{R}^s$ be an absolutely continuous function such that $\nabla\psi(\theta)$ is continuous at $\theta_0$. For all $n$, let all distributions $P_\theta$ in the family $\{P_\theta\}_{\theta \in \mathbb{R}^d}$ have density $p_\theta$ which satisfies $\int_{\mathbb{R}^d} \|\nabla p_\theta(x)\|_2^2 \, dx < \infty$. If $\lim_{c \to \infty} \lim_{n \to \infty} \sup_{\|h\|_2 < 1} \mathcal{I}^n(\theta_0 + ch/\sqrt{n})/n$ exists almost surely and is positive definite, denote it by $\rho$. Then for all sequences $(\hat{\theta}_n)_{n \geq 1}$ of statistics $S_n : \mathcal{X}^n \to \mathbb{R}^s$ and for all $u \in \mathbb{R}^s$*

$$\liminf_{c \to \infty} \liminf_{n \to \infty} \sup_{\|h\| < 1} \mathbb{E}_{\theta_0 + \frac{ch}{\sqrt{n}}}^n \left[ \left\langle \sqrt{n}\left( \hat{\theta}_n - \psi\left( \theta_0 + \frac{ch}{\sqrt{n}} \right) \right), u \right\rangle^2 \right] \geq u^\top \nabla\psi(\theta_0)^\top \rho^{-1} \nabla\psi(\theta_0) u$$

*Proof.* The main difference between our version of the proof and the one presented in Theorem 4 of Gassiat [10] is that we do not assume $\mathcal{I}^n = n\mathcal{I}$. We also select $\ell(x) = \langle u, x \rangle^2$ in particular. All the steps and notation remain the same except with $n\mathcal{I}$ replaced with $\mathcal{I}^n$ up until equation (13), which we define with a modified choice of $\Gamma_{c,n}$

$$\Gamma_{c,n} := \left( \int_{\mathcal{B}_p([0],1)} \nabla\psi(\theta_0 + ch/\sqrt{n})q(h)dh \right)^\top \left( \frac{1}{c^2}\mathcal{I}_q + \frac{1}{n}\int_{\mathcal{B}_p([0],1)} \mathcal{I}^n(\theta_0 + ch/\sqrt{n})q(h)dh \right)^{-1}$$

$$\times \left( \int_{\mathcal{B}_p([0],1)} \nabla \psi(\theta_0 + ch/\sqrt{n}) q(h) dh \right).$$

By definition of $\rho$, with probability 1,

$$\lim_{c \to \infty} \lim_{n \to \infty} \Gamma_{c,n} = \nabla \psi(\theta_0)^\top \rho^{-1} \nabla \psi(\theta_0)$$

$\square$

### D.1 Proof of Theorem 5.1

We will apply Corollary D.1 and apply it to two different choices of $\psi$ to get the full feature minimax bound and missing feature minimax bound respectively. For notational simplicty, let $P_\theta$ denote the distribution over $\{\tilde{\theta}_i^n\}_{i \in [m]}$ induced by $\theta$. $P_\theta$ is in the exponential family, so the conditions of Corollary D.1 are satisfied.

We begin by computing the Fisher Information. Let $P_\theta^j$ for $j \in [m]$ denote the distribution over $\tilde{\theta}_j^n \in \mathbb{R}^{d_j}$. Let $\mathcal{I}_i^n(\theta)$ denote the Fisher Information of $P_\theta^i$, and let $\mathcal{I}^n(\theta) = \sum_{i=1}^m \mathcal{I}_i^n$ denote the Fisher Information of $P_\theta$. Let $x_{i+}$ denote an arbitrary row of $X_{i+}$. Let $x_{i-}$ be drawn from $\mathsf{N}(\mu_{i-}(x_{i+}), \Gamma_{i-})$. Some straightforward calculations tell us $\mu_{i-}(x_{i+}) = \Sigma_{i\mp}\Sigma_{i+}^{-1}x_{i+}$ and $\Gamma_{i-} = \Sigma_{i-} - \Sigma_{i\mp}\Sigma_{i+}^{-1}\Sigma_{i\pm}$. From this we can deduce that $\theta_{i-}^T x_{i-}$ is distributed as $\mathsf{N}(\mu_{i-}^T\theta_{i-}, \theta_{i-}^T\Gamma_{i-}\theta_{i-})$; we use $\mu_{i-}$ in place of $\mu_{i-}(x_{i-})$ for simplicity. And $y_i$ is distributed as $P_\theta^i$ which is $\mathsf{N}(\theta^T\gamma, \theta_{i-}^T\Gamma_{i-}\theta_{i-} + \sigma^2)$ where $\gamma := [x_{i+}^T\Pi_{i+}, \mu_{i-}^T\Pi_{i-}]^T$. From this we can deduce that $P_\theta^i$ is $\mathsf{N}\left(J_i\Pi_i\theta, \beta_i^{-1}\widehat{\Sigma}_{i+}^{-1}\right)$, where $\beta_i^{-1} := \frac{\theta_{i-}\Gamma_{i-}\theta_{i-} + \sigma^2}{n}$; let $p_\theta^i$ denote its density. We know $\mathcal{I}^n(\theta) = \sum_{i=1}^m \mathcal{I}_i^n(\theta)$ due to independence. All that remains is to compute $\mathcal{I}_i^n(\theta)$.

$$\mathcal{I}_i^n(\theta) = \int \nabla_\theta \log p_\theta^i(z)[\nabla_\theta \log p_\theta^i(z)]^T p_\theta^i(z) dz.$$

We know that for some constant $C$,

$$\log p_i^\theta(z) = C + \frac{d_i}{2}\log(\beta_i) - \frac{\beta_i}{2}\left\| \widehat{\Sigma}_{i+}^{\frac{1}{2}}\theta_{i+} + \widehat{\Sigma}_{i+}^{\frac{1}{2}}\Sigma_{i+}^{-1}\Sigma_{i\pm}\theta_{i-} - \widehat{\Sigma}_{i+}^{\frac{1}{2}}z \right\|_2^2.$$

Taking derivaties we get that

$$\nabla_{\theta_{i+}} \log p_i^\theta(z) = -\beta_i\left[ \widehat{\Sigma}_{i+}\theta_{i+} + \widehat{\Sigma}_{i+}\Sigma_{i+}^{-1}\Sigma_{i\pm}\theta_{i-} - \widehat{\Sigma}_{i+}z \right]$$

$$\nabla_{\theta_{i-}} \log p_i^\theta(z) = \left[ -\frac{d_i}{n} + \left\| \widehat{\Sigma}_{i+}^{\frac{1}{2}}\theta_{i+} + \widehat{\Sigma}_{i+}^{\frac{1}{2}}\Sigma_{i+}^{-1}\Sigma_{i\pm}\theta_{i-} - \widehat{\Sigma}_{i+}^{\frac{1}{2}}z \right\|_2^2 \right]\beta_i\Gamma_{i-}\theta_{i-}$$
$$+ \left[ \Sigma_{i\mp}\Sigma_{i+}^{-1}\widehat{\Sigma}_{i+}\theta_{i+} + \Sigma_{i\mp}\Sigma_{i+}^{-1}\widehat{\Sigma}_{i+}\Sigma_{i+}^{-1}\Sigma_{i\pm}\theta_{i-} - \Sigma_{i\mp}\Sigma_{i+}^{-1}\widehat{\Sigma}_{i+}z \right]\beta_i$$

Let $b^2 = \left\| \widehat{\Sigma}_{i+}^{\frac{1}{2}}\theta_{i+} + \widehat{\Sigma}_{i+}^{\frac{1}{2}}\Sigma_{i+}^{-1}\Sigma_{i\pm}\theta_{i-} - \widehat{\Sigma}_{i+}^{\frac{1}{2}}z \right\|_2^2$. Now we compute the expectation over outer products:

$$\mathbb{E}[\nabla_{\theta_{i+}} \log p_i^\theta(z)\nabla_{\theta_{i+}} \log p_i^\theta(z)^T] = \beta_i\widehat{\Sigma}_{i+}$$

$$\mathbb{E}[\nabla_{\theta_{i+}} \log p_i^\theta(z)\nabla_{\theta_{i-}} \log p_i^\theta(z)^T] = \beta_i^2\widehat{\Sigma}_{i+}\beta_i^{-1}\widehat{\Sigma}_{i+}^{-1}\widehat{\Sigma}_{i+}\Sigma_{i+}^{-1}\Sigma_{i\pm} = \beta_i\widehat{\Sigma}_{i+}\Sigma_{i+}^{-1}\Sigma_{i\pm}$$

$$\mathbb{E}[\nabla_{\theta_{i-}} \log p_i^\theta(z)\nabla_{\theta_{i-}} \log p_i^\theta(z)^T] = \beta_i\Sigma_{i\mp}\Sigma_{i+}^{-1}\widehat{\Sigma}_{i+}\Sigma_{i+}^{-1}\Sigma_{i\pm}$$
$$+ \left( \frac{d_i^2}{n^2} + \mathbb{E}[b^2]\frac{2d_i}{n} + \mathbb{E}[b^4] \right)\beta_i^2\Gamma_{i-}\theta_{i-}\theta_{i-}^T\Gamma_{i-}$$
$$= \beta_i\Sigma_{i\mp}\Sigma_{i+}^{-1}\widehat{\Sigma}_{i+}\Sigma_{i+}^{-1}\Sigma_{i\pm} + \left( \frac{d_i^2}{n^2} + \frac{2\beta_i^{-1}d_i^2}{n} + \beta_i^{-2}(2d_i + d_i^2) \right)\beta_i^2\Gamma_{i-}\theta_{i-}\theta_{i-}^T\Gamma_{i-}$$

$$\mathcal{I}_i^n(\theta) = \int \nabla_\theta \log p_\theta^i(z)[\nabla_\theta \log p_\theta^i(z)]^T p_\theta^i(z) dz$$

$$= \int \Pi_i^T \begin{bmatrix} \nabla_{\theta_{i+}} \log p_\theta^i(z) \\ \nabla_{\theta_{i-}} \log p_\theta^i(z) \end{bmatrix} \begin{bmatrix} \nabla_{\theta_{i+}} \log p_\theta^i(z)^T & \nabla_{\theta_{i-}} \log p_\theta^i(z)^T \end{bmatrix} \Pi_i p_\theta^i(z) dz$$

$$= \frac{n}{\sigma^2 + \theta_{i-}^T \Gamma \theta_{i-}} \Pi_i^T \begin{bmatrix} \widehat{\Sigma}_{i+} & \widehat{\Sigma}_{i+} \Sigma_{i+}^{-1} \Sigma_{i\pm} \\ \Sigma_{i\mp} \Sigma_{i+}^{-1} \widehat{\Sigma}_{i+} & \Sigma_{i\mp} \Sigma_{i+}^{-1} \widehat{\Sigma}_{i+} \Sigma_{i+}^{-1} \Sigma_{i\pm} \end{bmatrix} \Pi_i$$

$$+ \Pi_i^T \begin{bmatrix} 0 & 0 \\ 0 & \left( \frac{d_i^2 \beta_i^2}{n^2} + \frac{2\beta_i d_i^2}{n} + 2d_i + d_i^2 \right) \Gamma_{i-} \theta_{i-} \theta_{i-}^T \Gamma_{i-} \end{bmatrix} \Pi_i$$

$$= \frac{n}{\sigma^2 + \theta_{i-}^T \Gamma \theta_{i-}} (Q_i + o_n(1))$$

The $o_n(1)$ term is due to strong law of large numbers. From this we know that, with probability 1,

$$\lim_{c \to \infty} \lim_{n \to \infty} \sup_{\|h\|_2 < 1} \frac{\mathcal{I}^n(\theta_0 + ch/\sqrt{n})}{n} = \sum_{i=1}^m \frac{1}{\sigma^2 + \theta_{i-}^T \Gamma \theta_{i-}} Q_i =: \rho$$

Applying Corollary D.1 with $\psi \mathbb{R}^d \to \mathbb{R}^d$ as the identity function $\psi(x) = x$ gives the full-feature minimax lower bound. Applying Corollary D.1 with $\psi \mathbb{R}^d \to \mathbb{R}^{d_i}$ as $\psi(x) = T_i x$ gives the missing-feature minimax lower bound.

### D.2 Proof of Theorem 5.2

We will apply Corollary D.1 and apply it to two different choices of $\psi$ to get the full feature minimax bound and missing feature minimax bound respectively. For notational simplicity, we will use $P_\theta$ in place of $P_\theta^y$. $P_\theta$ is in the exponential family, so the conditions of Corollary D.1 are satisfied.

We begin by computing the Fisher Information. Let $P_\theta^j$ for $j \in [m]$ denote the distribution over $y_j \in \mathbb{R}^n$. Let $\mathcal{I}_i^n(\theta)$ denote the Fisher Information of $\mathbb{P}_\theta^i$, and let $\mathcal{I}^n(\theta) = \sum_{i=1}^m \mathcal{I}_i^n(\theta)$ denote the Fisher Information of $P_\theta$.

Let $x_i^{(k)}, y_i^{(k)}$ be the $k$th sample from agent $i$. We will let $\mathcal{I}_i^{(k)}(\theta)$ be the fisher information of $y_i^{(k)}$. We know that $\mathcal{I}_i^n(\theta) = \sum_{k=1}^n \mathcal{I}_i^{(k)}(\theta)$ by independence. Some straightforward calculations tell us that $x_{i-}^{(k)}$ is distributed as $\mathsf{N}(\mu, \Gamma)$ where $\mu = \Sigma_{i\mp} \Sigma_{i+}^{-1} x_{i+}^{(k)}$ and $\Gamma = \Sigma_{i-} - \Sigma_{i\mp} \Sigma_{i+}^{-1} \Sigma_{i\pm}$. From this we can deduce that $\theta_{i-}^T x_{i-}^{(k)}$ is distributed as $\mathsf{N}(\mu^T \theta_{i-}, \theta_{i-}^T \Gamma \theta_{i-})$. And $y_i^{(k)}$ is distributed as $\mathsf{N}(\theta^T \gamma, \theta_{i-}^T \Gamma \theta_{i-} + \sigma^2)$ where $\gamma := \Pi_{i+}^T x_{i+}^{(k)} + \Pi_{i-}^T \mu$.

Let $\phi := \frac{z - \gamma^T \theta}{\sigma^2 + \theta_{i-}^T \Gamma \theta_{i-}}$ and $\Delta := \phi^2 - \frac{1}{\sigma^2 + \theta_{i-}^T \Gamma \theta_{i-}}$. Using $p_\theta^{ik}$ denote the density of $x_{i-}^{(k)}, y_i^{(k)}$, we can calculate the derivative of the log density

$$\nabla_{\theta_{i+}} \log p_\theta^{ik}(z) = \frac{z - \theta_{i+}^T x_{i+}^{(k)} - \theta_{i-}^T \mu}{\sigma^2 + \theta_{i-} \Gamma \theta_{i-}} x_{i+}^{(k)} = \phi x_{i+}^{(k)}$$

$$\nabla_{\theta_{i-}} \log p_\theta^{ik}(z) = \Delta \Gamma \theta_{i-} + \phi \mu.$$

Using the facts that $\mathbb{E}[\phi] = 0$, $\mathbb{E}[\phi^2] = \frac{1}{\sigma^2 + \theta_{i-} \Gamma \theta_{i-}}$, $\mathbb{E}[\phi\Delta] = 0$, and $\mathbb{E}[\Delta^2] = \frac{2}{(\sigma^2 + \theta_{i-} \Gamma \theta_{i-})^2}$, where the expectation is an integral over $z$, we have that

$$\mathcal{I}_i^{(k)}(\theta) = \int \nabla_\theta \log p_\theta^{ik}(z) [\nabla_\theta \log p_\theta^{ik}(z)]^T p_\theta^{ik}(z) dz$$

$$= \int \Pi_i^T \begin{bmatrix} \nabla_{\theta_{i+}} \log p_\theta^{ik}(z) \\ \nabla_{\theta_{i-}} \log p_\theta^{ik}(z) \end{bmatrix} \begin{bmatrix} \nabla_{\theta_{i+}} \log p_\theta^{ik}(z)^T & \nabla_{\theta_{i-}} \log p_\theta^{ik}(z)^T \end{bmatrix} \Pi_i p_\theta^{ik}(z) dz$$

$$= \Pi_i^T \begin{bmatrix} \mathbb{E}[\phi^2] x_{i+}^{(k)} (x_{i+}^{(k)})^T & \mathbb{E}[\phi x_{i+}^{(k)} (\Delta \Gamma \theta_{i-} + \phi\mu)^T] \\ \mathbb{E}[(\Delta \Gamma \theta_{i-} + \phi\mu)(\phi x_{i+}^{(k)})^T] & \mathbb{E}[(\Delta \Gamma \theta_{i-} + \phi\mu)(\Delta \Gamma \theta_{i-} + \phi\mu)^T] \end{bmatrix} \Pi_i$$

$$= \frac{1}{\sigma^2 + \theta_{i-} \Gamma \theta_{i-}} \Pi_i^T \begin{bmatrix} x_{i+}^{(k)} (x_{i+}^{(k)})^T & x_{i+}^{(k)} \mu^T \\ \mu (x_{i+}^{(k)})^T & \mu\mu^T + \frac{2}{\sigma^2 + \theta_{i-}^T \Gamma \theta_{i-}} \Gamma \theta_{i-} \theta_{i-}^T \Gamma \end{bmatrix} \Pi_i$$

$$= \frac{1}{\sigma^2 + \theta_{i-}\Gamma\theta_{i-}}\Pi_i^T \begin{bmatrix} x_{i+}^{(k)}(x_{i+}^{(k)})^T & x_{i+}^{(k)}(x_{i+}^{(k)})^T\Sigma_{i+}^{-1}\Sigma_{i\pm} \\ \Sigma_{i\mp}\Sigma_{i+}^{-1}x_{i+}^{(k)}(x_{i+}^{(k)})^T & \Sigma_{i\mp}\Sigma_{i+}^{-1}x_{i+}^{(k)}(x_{i+}^{(k)})^T\Sigma_{i+}^{-1}\Sigma_{i\pm} + \frac{2}{\sigma^2+\theta_{i-}^T\Gamma\theta_{i-}}\Gamma\theta_{i-}\theta_{i-}^T\Gamma \end{bmatrix} \Pi_i.$$

From this we can sum over

$$\mathcal{I}_i^n(\theta) = \sum_{k=1}^n \mathcal{I}_i^{(k)}(\theta)$$

$$= \frac{n}{\sigma^2 + \theta_{i-}^T\Gamma\theta_{i-}}\Pi_i^T \begin{bmatrix} \widehat{\Sigma}_{i+} & \widehat{\Sigma}_{i+}\Sigma_{i+}^{-1}\Sigma_{i\pm} \\ \Sigma_{i\mp}\Sigma_{i+}^{-1}\widehat{\Sigma}_{i+} & \Sigma_{i\mp}\Sigma_{i+}^{-1}\widehat{\Sigma}_{i+}\Sigma_{i+}^{-1}\Sigma_{i\pm} + \frac{2}{\sigma^2+\theta_{i-}^T\Gamma\theta_{i-}}\Gamma\theta_{i-}\theta_{i-}^T\Gamma \end{bmatrix} \Pi_i$$

$$= \frac{n}{\sigma^2 + \theta_{i-}^T\Gamma\theta_{i-}}\left( Q_i + o_n(1) + \Pi_i^T \begin{bmatrix} 0 & 0 \\ 0 & \frac{2}{\sigma^2+\theta_{i-}^T\Gamma\theta_{i-}}\Gamma\theta_{i-}\theta_{i-}^T\Gamma \end{bmatrix} \Pi_i \right)$$

The $o_n(1)$ term is due to strong law of large numbers. From this we know that, with probability 1

$$\lim_{c\to\infty}\lim_{n\to\infty}\sup_{\|h\|_2<1} \frac{\mathcal{I}^n(\theta_0 + ch/\sqrt{n})}{n}$$

$$= \sum_{i=1}^m \frac{1}{\sigma^2 + \theta_{i-}^T\Gamma\theta_{i-}}\left( Q_i + \Pi_i^T \begin{bmatrix} 0 & 0 \\ 0 & \frac{2}{\sigma^2+\theta_{i-}^T\Gamma\theta_{i-}}\Gamma\theta_{i-}\theta_{i-}^T\Gamma \end{bmatrix} \Pi_i \right) =: \rho$$

Applying Corollary D.1 with $\psi\mathbb{R}^d \to \mathbb{R}^d$ as the identity function $\psi(x) = x$ gives the full-feature minimax lower bound. Applying Corollary D.1 with $\psi\mathbb{R}^d \to \mathbb{R}^{d_i}$ as $\psi(x) = T_i x$ gives the missing-feature minimax lower bound.

One final transformation remains to get the form of this lower bound to match the one in the theorem statement. We know that from Cauchy-Schwartz that for all $u \in \mathbb{R}^{d-d_i}$

$$\frac{u^T\Gamma\theta_{i-}\theta_{i-}^T\Gamma u}{\theta_{i-}^T\Gamma\theta_{i-}} = \frac{(u^T\Gamma^{\frac{1}{2}}\Gamma^{\frac{1}{2}}\theta_{i-})^2}{\theta_{i-}^T\Gamma\theta_{i-}} \leq u^T\Gamma u.$$

Using this fact and the definition of $\Gamma$ and $Q_i$ we have that

$$\frac{1}{\sigma^2 + \theta_{i-}^T\Gamma\theta_{i-}}\left( Q_i + \Pi_i^T \begin{bmatrix} 0 & 0 \\ 0 & \frac{2}{\sigma^2+\theta_{i-}^T\Gamma\theta_{i-}}\Gamma\theta_{i-}\theta_{i-}^T\Gamma \end{bmatrix} \Pi_i \right) \preceq \frac{2n}{\sigma^2 + \theta_{i-}^T\Gamma\theta_{i-}}\Sigma.$$

Using this bound gives our final result.

### D.3 Proof of Corollary 5.3

The existence of the limits is a consequence of strong law of large numbers. To further show the inequality in the limit, we note that

$$\frac{1}{m}\sum_{i=1}^m \left( \frac{\Sigma}{\sigma^2 + \theta_{i-}^T\Gamma_{i-}\theta_{i-}} - \frac{T_i^\top\Sigma_{i+}T_i}{\sigma^2 + \theta_{i-}^T\Gamma_{i-}\theta_{i-}} \right) = \frac{1}{m}\sum_{i=1}^m \frac{\Pi_i^\top \begin{bmatrix} 0 & 0 \\ 0 & \Gamma_{i-} \end{bmatrix} \Pi_i}{\sigma^2 + \theta_{i-}^T\Gamma_{i-}\theta_{i-}}$$

$$\preceq \frac{1}{m}\sum_{i=1}^m \frac{\Pi_i^\top \begin{bmatrix} 0 & 0 \\ 0 & \Gamma_{i-} \end{bmatrix} \Pi_i}{\sigma^2} \preceq \frac{1}{m}\sum_{i=1}^m \frac{\Pi_i^\top \begin{bmatrix} 0 & 0 \\ 0 & \Sigma_{i-} \end{bmatrix} \Pi_i}{\sigma^2} \to \frac{p\,\mathrm{diag}(\Sigma) + p^2(\Sigma - \mathrm{diag}(\Sigma))}{\sigma^2},$$

where the last step holds with probability one by strong law of large numbers. This is true as by our random missing model, $\Sigma_{ij}$ is not observed with probability $p$ if $i = j$, and $p^2$ if $i \neq j$. We can further derive that

$$\frac{p\,\mathrm{diag}(\Sigma) + p^2(\Sigma - \mathrm{diag}(\Sigma))}{\sigma^2} \preceq \frac{p\lambda_1(\Sigma)I}{\sigma^2} \preceq \frac{p\kappa\Sigma}{\sigma^2} \preceq \frac{p\lambda_1(\Sigma)I}{\sigma^2}$$

$$\overset{(i)}{\preceq} \frac{p\kappa(\sigma^2 + \|\theta\|_\Sigma^2)}{\sigma^2}\frac{1}{m}\sum_{i=1}^m \frac{\Sigma}{\sigma^2 + \theta_{i-}^T\Gamma_{i-}\theta_{i-}}.$$

In (i), we make use of the fact that

$$\Sigma \succeq \Pi_i^\top \begin{bmatrix} 0 & 0 \\ 0 & \Gamma_{i-} \end{bmatrix} \Pi_i$$

and therefore $\|\theta\|_\Sigma^2 \geq \|\theta_{i-}\|_{\Gamma_{i-}}^2$. By our choice of $p \leq \frac{1}{2}\kappa^{-1}(1 + \|\theta\|_\Sigma^2/\sigma^2)^{-1}$, we can conclude that

$$\lim_{m\to\infty} \frac{1}{m} \sum_{i=1}^m \frac{T_i^\top \Sigma_{i+} T_i}{\sigma^2 + \theta_{i-}^T \Gamma_{i-} \theta_{i-}} \succeq \lim_{m\to\infty} \frac{1}{m} \sum_{i=1}^m \frac{\Sigma/2}{\sigma^2 + \theta_{i-}^T \Gamma_{i-} \theta_{i-}}.$$