# OpenReview forum: "Collaboratively Learning Linear Models with Structured Missing Data"
_NeurIPS.cc/2023/Conference — NeurIPS 2023 poster_

### Official Review · Reviewer_93Dr · 2023-06-19

**Soundness:** 3 good
**Presentation:** 2 fair
**Contribution:** 3 good
**Rating:** 5
**Confidence:** 3

**Summary:**

The paper discusses the idea of estimating least squares collaboratively when each agent has access to a different set of features of the same data. The aim is to design an effective and efficient algorithm in terms of communication cost (various agents transferring/communicating information/data). The paper introduces a new algorithm *COLLAB* that is efficient and also applicable in security fields where features can not be transferred between agents/sensors/machines/systems. The authors theoretically prove that the proposed algorithm *COLLAB* is minimax optimal and perform a set of experiments to showcase the capability of the proposed algorithm *COLLAB*.

**Strengths:**

The paper discusses the setup typical in many fields (especially security applications) where data can not be transferred between agents due to security reasons or input-output constraints (network bottleneck). In the setup discussed, when the agent has a linear model, ordinary least squares (OLS) is a logical way to solve and get the parameter value. The idea behind *COLLAB* is logical and intuitive however needs more clarity in the presentation. The comparison against the imputation methods is also logical, as these are go-to models for such setups. The authors derive local minimax lower bounds to showcase that the proposed algorithm *COLLAB* is very close to the optimal, which is interesting. For the correctness of this section, I would rely on other reviewers as I was not able to follow the derivations clearly. The experiments are also a good mixture of real-world and synthetic data sets where the capability of the proposed algorithm is shown.

**Weaknesses:**

* A nice idea, but some assumptions are very hard/limited, and the evaluations are limited.
* Some details (especially Sections 3.1 and 3.2) are presented convolutedly and are hard to grab. Some minor clarifications:
  - L110, is the `x` written same as L99? It probably should be `X`?
  - L132, L133: I am still not sure what `(i)` refers to here.
  - In general, I believe it is convenient to stick to the notation where bold **`X`** is a matrix, bold **`x`** is a vector and `x` is a scalar. It makes it easy to follow the equations.
* Evaluations are limited. More empirical evaluations are to be performed to showcase the results of the proposed algorithm against other methods. A limitation here is the linear model in the agents, which limits the modelling capability of the model.
* Considering non-linear cases as well would make the paper more solid. Then the agents are more flexible, and the authors can experiment with more complex data sets.
* Computation benefits should be shown empirically as well. Currently, it is only shown theoretically.


**Questions:**

* The paper assumes that all the agents have the same linear model. How restrictive is it? One of the agents has a $R^{16}$ dimension feature vector, whereas the other might have a $R^2$ dimension feature vector. So, the same complex model does not seem logical here.
* Second assumption is that the agent has enough data to estimate $\Sigma$. How will it impact the algorithm if there is little data and $\Sigma$ is unreliable/biased?
* A toy example to plot various parameter $\theta_i$ of the agents, how they combine to give $\theta_{global}$, and how they compare with the oracle value of $\theta$.
* A setup where the computational benefits can be shown empirically as well?


**Limitations:**

The authors do not discuss them, and I do not see a limitation directly of this work.

---

> ### Author Rebuttal · Authors · 2023-08-07
>
> Thank you for your review.
>
> **Response to W1: limited assumptions and evaluations**: We admit that our current theory relies heavily on Gaussianity assumption.These generalizations have various technical challenges, which make giving strong theoretical guarantees (i.e., what we were able to do in our linear Gaussian model) very difficult. In our work, we opt to provide these strong guarantees in lieu of more generality. We point out that the experiments are on real data, suggesting potential generalization of our theory beyond the Gaussian case. Removing linearity/Gaussianity would be an interesting future direction that we have discussed in Section 7. Having said that, we are happy to  explain in more detail if you want further clarification on  particular assumptions/evaluations.
>
> **Response to W2: minor clarifications**: Thank you for the suggestions, we will make clarifications in the camera-ready version for the following points.
> 2.1: Yes, the $x\in \mathbb{R}^d$ in line 110 is the same as the $x\in\mathbb{R}^d$ in line 99.
> 2.2: The $(i)$ refers to the second equality in the equation between lines 131 and 132.
> 2.3: This is a good suggestion, and we will try to make the notation more clear in the camera-ready version.
>
> **Response to W3: limited evaluations**: We partially respond in Response to weakness 1. We want to emphasize that we believe our main contribution is theory. The focus of our preliminary experiments is to show that our method does not overfit to the Gaussian data setting.
>
> **Response to W4: non-linear cases**:  We respond in Response to weakness 1.
>
> **Response to W5: empirical justification of computational benefits**: Thank you for bringing this up.e think there is some confusion, as our focus is on communication cost instead of computational cost. Please correct us if we misunderstood. We note that the communication costs in Table 1 are actually not asymptotic, local imputation requires communicating $d^2$ real numbers, global imputation $nd_i$ real numbers, and our method $d_i^2$ real numbers; we will clarify this in the camera-ready version. For this reason, we believe benchmarking real communication costs is not necessary.
>
> **Response to Q1: dimensional differences among agents**: Indeed, our model setup assumes the same underlying model with agent-dependent, unobserved dimensions. We want to clarify that it is not restrictive to dimensional differences among agents. Our setup allows one agent to observe data in $\mathbb{R}^{16}$ dimensions while another observes data in $\mathbb{R}^{2}$ dimensions; the second agent would just observe fewer dimensions than the first agent. It would be very challenging to develop theory when the underlying models are different for each agent, as there is no shared global model we can analyze. We feel this is beyond the scope of the discussion in our paper.
>
> **Response to Q2: unreliable/biased estimate of the covariance**: This is an interesting point. We first want to point out a potential confusion that could be caused by a typo in the definition of $\hat{W}\_i^g$ (line 155). The numerator is supposed to be the sample sub-covariance $\hat{\Sigma}\_{i+} = X\_{i+}^\top X\_{i+}/n$ instead of the exact sub-covariance matrix $\Sigma\_{i+}$. In fact, we use $\hat{\Sigma}\_{i+}$ in Algorithm 1 and in the proof of Corollary 3.2 in the submitted supplementary materials. It is not clear what is the optimal procedure if we do not have a consistent estimate for the population covariance, which essentially boils down to the harder problem of distributional shift. This could be a future direction and we will include it in our discussion.
>
> **Response to Q3: suggestion of a toy example**: Thank you for this experimental suggestion. As $\theta_i$ are in general high dimensional (>3), the plots we can visually make would probably be $\ell\_2$ error against the ground-truth, which might be less clear, so we opted for doing real data experiments due to page-limit. Please let us know if you have any suggestions of how to visualize a toy example.
>
> **Response to Q4: computational benefits**: See our response to W5.

---

> > ### Comment · Reviewer_93Dr · 2023-08-14
> >
> > Thank you for the response!
> >
> > After going through other reviews and replies, I would stay with my original score.
> >
> > In my opinion, a discussion on current assumptions and how they can be relaxed, agents with different models as agents can have different model complexity, communication cost, and a simple example to explain the benefits of the proposed model intuitively would be a good addition and make the paper complete.

---

### Official Review · Reviewer_N12y · 2023-07-06

**Soundness:** 4 excellent
**Presentation:** 4 excellent
**Contribution:** 2 fair
**Rating:** 6
**Confidence:** 3

**Summary:**

The authors investigate collaborative learning of least squares estimates for multiple agents with varying feature subsets. The goal is to coordinate the agents efficiently to achieve optimal estimators without exchanging labeled data. To address this, the authors propose the distributed algorithm Collab, consisting of local training, aggregation, and distribution steps. Despite not sharing labeled data, Collab approaches near-asymptotic local minimax optimality, outperforming methods that do utilize labeled data. They validate our approach through experiments on real and synthetic datasets.

**Strengths:**

Distributed learning with heterogeneous data sources is a problem of broad interest. In this study, the authors tackle this problem in a simplified setting and provide robust theoretical guarantees. Their theoretical results are strong, demonstrating the solid foundation of their approach. In addition, the point of minimizing communication resources is an interesting angle. Furthermore, the experimental results are compelling, further supporting the effectiveness of their method.

**Weaknesses:**

- Settings need more justification: the authors discuss a setting where a linear regression problem is running on satellites with completely different features. This seems restrictive and can the authors elaborate more on the motivation of their study?
- Random X with zero mean. If X is not zero-mean, then the regression with partial information is not consistent anymore (different features may have correlations). I can imagine this is a very common scenario and can the authors provide some discussions on this?


**Questions:**

In Section 7, is that the numerator in *generalizing to non-linear models* requires sharing global information for $x_{i+}$? What are the possible ways to alleviate this constraint?

**Limitations:**

I do not foresee any potential societal impact of this work.

---

> ### Author Rebuttal · Authors · 2023-08-07
>
> Thank you for your review.
>
> **Response to justifications for our setting**: Regarding the satellite application, we agree it is a stylized example and have expanded the introduction to discuss other potential applications like sensor networks, weather stations, and hospitals. The key aspects we wish to capture are: 1) agents have different features due to heterogeneous data sources, 2) agents wish to leverage correlations and train collaborative models, 3) communication constraints exist. We believe these aspects apply broadly. The high level motivation for our work is to study how practitioners should handle hardware heterogeneity in communication-constrained collaborative learning environments. Works in federated learning have thought about how to do training with heterogeneous computational hardware; e.g., learning with client phones that have different processing power [Yang et al.]. But little attention has been focused on settings on heterogeneous measurement devices; e.g., what would we do if the features collected by each agent were heterogeneous? As we show, this problem is challenging, even in the simplest linear regression setting.
>
> **Response to zero-mean data**: Thank you for bringing up the point about zero-mean assumption. It is standard practice in machine learning to first center the data by subtracting the mean. This pre-processing step results in zero-mean features. This is why many other theory papers study the zero mean setting (e.g., Hastie et al., [3]). In fact, we followed this approach in the real census data experiment by centering each feature before model training and evaluation. As seen in Figure 1, COLLAB either outperforms or is competitive against baselines, indicating it is robust even when the zero-mean assumption is violated in practice.
>
> **Response to the question**: This is a good question. We want to clarify that the formula is under the expectation of $x\_{i+}$ which is a population quantity that each agent can estimate consistently and locally. So, no, the numerator does not require sharing individual data samples $x\_{i+}$, as we assume that each agent has $n$ labeled samples with $n$ growing to infinity (over the subset of the features they can observe). Thus, each agent can estimate the numerator individually with the samples they have access to and with their own local model.
>
> **Additional References**
>
> Hastie, Trevor J. et al. “Surprises in High-Dimensional Ridgeless Least Squares Interpolation.” Annals of statistics 50 2 (2019): 949-986 .
>
> Yang, Chengxu et al. “Characterizing Impacts of Heterogeneity in Federated Learning upon Large-Scale Smartphone Data.” Proceedings of the Web Conference 2021 (2021): n. pag.

---

> > ### Comment · Reviewer_N12y · 2023-08-12
> >
> > For the "response to the question": thanks for the answer!
> >
> > This potentially inspires a practical iterative algorithm for non-linear settings (e.g., $\theta$ is the parameter for a complex neural network): to solve $\theta$ globally, at each step the global processor sends the current $\theta$ to the local processors, each local processor then computes the gradient (or a sub-gradient) of $\theta$ for the proposed local loss function and return to the global processor. Based on the gradient information, the global processor then updates its $\theta$ accordingly. This procedure can be conducted iteratively. One particular choice for $f(x_i^{+}; T_{i}\theta)$ can be $f([0, 0, 0, ..., x_{i}^{+}, ..., 0, 0, 0]; \theta)$ (taking the inputs for other features as 0 for the local processor $i$ when computing the prediction given the global $\theta$, commonly used for deep learning when some features are missing).
> >
> > This can be viewed as a generalization of existing federated learning algorithms to the heterogenous feature-observation setting. I hope the authors can elaborate on this in their revised version as I believe this will likely make the impact of this work significantly larger.

---

> > > ### Author Response · Authors · 2023-08-14
> > >
> > > That is a good point. There could be possible generalizations of our work to heterogeneous feature-observation federated learning settings. The proposed loss could indeed be minimized in a distributed/federated manner to aggregate local models iteratively. If we understand you correctly, we believe the algorithm would look like
> > > 1. Send $\hat{\theta}$ to each agent. Have the agent minimize their own loss (initialized at $\hat{\theta}$), and call the final parameter $\hat{\theta}_i$.
> > > 2. Minimize the proposed loss function (the one between lines 324 and 325) in a federated way as a means of aggregating the model. Call the final parameter $\hat{\theta}$.
> > > 3. Repeat.
> > >
> > > Thank you for bringing up this connection to federated learning. We will add this discussion to the camera ready version.

---

### Official Review · Reviewer_vMLQ · 2023-07-09

**Soundness:** 2 fair
**Presentation:** 2 fair
**Contribution:** 2 fair
**Rating:** 4
**Confidence:** 5

**Summary:**

The paper studies statistical inference in learning a linear regression model in the cross-silo or vertical federated learning setting. The paper formulates the problem as a missing data problem and chooses single imputation methods to deal with the associated inference of the common parameter. The paper shows the theoretical properties of the proposed estimator under some assumptions. Finally the paper compares their results with other existing methods.

**Strengths:**

The paper studies a very important problem and relatively easy to follow.

**Weaknesses:**

1. The paper is not carefully written with confusing notations and problem formulation:

1(a). In Section 2, it says that “the $i^{th}$ agent has data $(x_{i+}, y)$” and then later uses $y_i$ to denote the $i^{th}$ agent labeled data. It is unclear whether these labels are the same or not. If they are the same, this is an unrealistic assumption in practice and obviously contrary to the common assumption in the literature that only one active party has access to the label data.

1(b). Equation (1) defines a weighted MSE, with expectation assessed with respect to the feature $x$, which is problematic. Why not consider the typical unweighted MSE with expectation evaluated with respect to $y$ in regression setting?

2. It seems that the contribution of the paper is to apply existing single imputation based missing data method to vertical federated learning setting, with the exception that here they assume each agent has access to its own label data (which again is problematic).

3. The paper does compare their methods with several other methods in the experiment. However, there are not enough discussions of these methods.

**Questions:**

What is the motivation to consider minimizing a weighted empirical loss in Equation (3)?

**Limitations:**

The paper considers single imputation, claiming that this is okay given that the goal is estimation error instead of confidence intervals. However, many theoretical results given in the paper are about the asymptotical distributions of the proposed estimators. Do these asymptotical distributions properly consider the uncertainty associated with the missing data imputation?

---

> ### Author Rebuttal · Authors · 2023-08-07
>
> Thank you for your review.
>
> **Based on your comments, we believe there is a misunderstanding**. We are not doing vertical federated learning [Liu et al.]. Unlike vertical federated learning, agents in our framework are not measuring the same underlying set of users. Vertical federated learning is not the right way to model settings like the satellite and seismic sensor estimation problem examples we discussed in the introduction, as this would mean, each agent (i.e., the sensor) would take (different) measurements of the same locations at the same time. In the setting we study, each agent has their own input data and labels. Thank you for pointing this out; we will clarify this in the camera ready version.
>
> **Response to 1a**: We believe this confusion stems from some notational issues. In line 102, $(x, y)$ refers to a single sample drawn from the generating distribution. In line 104, $y\_i\in \mathbb{R}^n$ refers to the vector of labels the agent observes. To be clear, as said in line 103, each agent gets $n$ draws of $(x, y)$ from the generating distribution and observes $X\_{i+}$ and $y\_i$: the labels are not the same across agents. Furthermore, just to make sure there is no confusion, the features observed by agent $i$ and agent $j$ ($j\neq i$) are *not* different subsets of the same $n$ feature vectors: each agent draws $n$ fresh covariates and observes a subset of the covariates. Finally, as we are not doing vertical federated learning, we do not have an “active party” constraint on the labeled data. We will make this very clear in the camera ready version.
>
> **Response to 1b**: We are confident that what we wrote is the standard notion of prediction/generalization error for a fresh sample $(x \in \mathbb{R}^d, y \in \mathbb{R})$. In the regression setting, weighted MSE with respect to the data distribution is the same up to an additive constant as the test error on a fresh sample: $\mathbb{E}\_{x, y}[(\langle x, \hat{\theta}\rangle) - y)^2] = \mathbb{E}\_x[(\langle x, \hat{\theta} \rangle - \langle x, \theta \rangle)^2] + \sigma^2 = \\|\hat{\theta} - \theta \\|\_{\Sigma}^2+ \sigma^2$. This formulation is standard (e.g., see section 2.1 in Hastie et al.).
>
> **Response to 2**: First, Collab is not doing single imputation at all. In fact, in section 4, we compare our method Collab against traditional single imputation methods. Second, we are not doing vertical federated learning as we discussed above.
>
> **Response to 3**: Can you provide more details about what discussion you feel is missing? Is there anything you want us to clarify? We are open to suggestions. We want to emphasize that while we chose to compare against imputation methods in Section 4, our method is asymptotically instance-optimal (as shown by our lower bounds), meaning that no algorithm could theoretically perform statistically better. In this sense, our lower-bound is the ultimate theoretical “baseline”.
>
> **Additional References**
>
> Hastie, Trevor J. et al. “Surprises in High-Dimensional Ridgeless Least Squares Interpolation.” Annals of statistics 50 2 (2019): 949-986 .
>
> Liu, Yang et al. “Vertical Federated Learning.” ArXiv abs/2211.12814 (2022): n. pag.

---

> > ### Comment · Reviewer_vMLQ · 2023-08-22
> >
> > Thank you for your reply. I've gone through the authors' rebuttal, and increased my score.

---

> ### Comment · Area_Chair_psaR · 2023-08-16
> **Please read the rebuttal and other reviews**
>
> Dear reviewer,
>
> The authors have posted a rebuttal. Please acknowledge that you have read it and indicate whether they have adequately addressed your concerns/comments. Your "strong reject" score indicates a significant technical flaw with the paper, and is in contradiction with the other scores on this paper. Please engage with the authors and clarify whether there is actually the technical flaw you're claiming. The author-reviewer discussion phase ends on Aug 21 so please discuss with the authors before that if you need any more clarifications.
> Thanks,
> AC

---

### Official Review · Reviewer_Mws6 · 2023-07-23

**Soundness:** 3 good
**Presentation:** 3 good
**Contribution:** 2 fair
**Rating:** 5
**Confidence:** 2

**Summary:**

Summary
-------

The paper studies collaborative linear regression when m agents attempt to collaboratively
estimate a linear model, under communication constraints. Each agent i only observes di
of the d features. A central server designs a protocol to elicit sufficient information
from each agent and compute the parameter theta of the linear model so as to minimize
communication costs and the estimation error.
The authors present, what appears to be a near-complete solution, for the case where the
covariates are distributed as a Gaussian and when the covariance matrix is known. This
includes asymptotic normality results for the proposed estimator and lower bounds which
match as n goes to infinity. The authors also compare their method against other baselines
based on imputing data and show that their estimation errors are no worse but at
significantly lower communication cost.

Decision: While the paper is well-presented and pleasant to read, I am not an expert in
this topic and did not have the time to go through the proofs in detail. As such,
I am unable to evalute the technical merit of the paper (challenges of the problem,
novelty of proof techniques). I have given a positive score with low confidence to reflect
this but will heed to more expert reviewers during the discussion.


Detailed comments
-----------------

The authors have used local and global imputation
baselines in Table 1 to show the communication benefits, and have gone on to show that
their method does no worse theoretically than these methods.
However, I am not sure if these are particularly strong baselines to compete with; for
instance, I would not have expected communicating all data points to be necessary.
In fact, at the outset, my intuition suggested a solution which computes local
coefficients by each agent which are then aggregated by the server in an appropriate
fashion.  It would have been helpful if the authors had better illustrated the challenges
in doing so. For instance, is the method in 3.1 the most natural way to solve this
problem, or are there other naive ways to aggregate the coefficients that yield
sub-optimal solutions?

The same applies to the non-Gaussian case and the setting when the co-variance matrices
are unknown. The authors could have done a better job of illustrating the challenges.
In general, I did not get a sense for how challenging this problem setting was to
appreciate the contributions by the authors.

Do the results in Theorems 3.1-3.2 implicitly capture the difficulty of the problem in
terms of the number of covariates each agent has access to?
- For instance, in the worst case there could be only n samples (when there is a perfect
    partition of the covariates among the agents, and they have the same points),
    but in the best case there could be mn
    samples (when all agents have all the covariates and have distinct points)
If so, is it possible to make this more explicit in the results.

The paper is largely well-written. The problem is well-motivated, the setting is described
clearly, and the method/results are organized well. I would have however liked to see a
sketch of the main proof ideas and how they differ from similar results in the linear
regression literature. A breakdown of the key proof challenges that the authors had to
overcome would have also been useful.

What was the reason for the discussion around o(n) communication complexity? It appears
that local imputation methods and the method of the authors are able to achieve consistent
estimation with communication cost that does not depend on n.

In the experiments, the imputation-based methods outperform the method of the authors when
there are more samples. Intuitively, I would have expected this since they are more
communication-heavy. However, this is not the case in the low-sample regime. Can you
explain why this is the case?


**Strengths:**

See above.

**Weaknesses:**

See above.

**Questions:**

See above.

**Limitations:**

See above.

---

> ### Author Rebuttal · Authors · 2023-08-06
>
> Thank you for your review.
>
> **Response to baseline strength**: Our method is asymptotically instance-optimal (shown by our lower bounds), meaning that no algorithm could perform statistically better on any specific problem instance. In some sense, our lower-bound is the ultimate theoretical “baseline”. The motivation for comparing baseline algorithms which are “stronger” than our method in terms of communication cost is to ultimately show in the experiments section that Collab is not overfit to the assumptions of our theory and performs well against communication-ignorant, conventionally-adopted methods, like imputation.
>
> **Response to the suggested intuitive solution**: If we understand correctly, our method falls in the scope of your intuition—Collab takes the locally computed coefficients and debiases them using covariance information. We note that we need covariance information from the agents, otherwise the debiasing procedure would return a biased estimate of the parameter. Having said this, we do want to point out that our aggregation approach is novel to the best of our knowledge and not standard in the missing data literature. As we discuss in the related work, imputation-type algorithms are more standard, which is also why we chose to baseline against them in our theory and experiments.
>
> **Response to non-Gaussian/unknown covariance case**: We first want to point out a potential confusion that could be caused by a typo in the definition of $\hat{W}\_i^g$ (line 155). The numerator is supposed to be the sample sub-covariance $\hat{\Sigma}\_{i+} = X\_{i+}^\top X\_{i+}/n$ instead of the exact sub-covariance matrix $\Sigma\_{i+}$. In fact, we use $\hat{\Sigma}\_{i+}$ in Algorithm 1 and in the proof of Corollary 3.2 in the submitted supplementary materials. We discuss the non-gaussian setting in lines 147-150 and section 7; to summarize, in the gaussian setting, we can estimate $W_i^\star$ from the data we have access to. In the non-gaussian setting we cannot. We are happy to answer any specific questions you may have about this.
>
> **Response to the implicit difficulty in Theorem 3.1 and Corollary 3.2**: Yes, you are right. Theorem 3.1 and Corollary 3.2 (and our lower bounds) capture the difficulty of the problem in terms of the number of covariates each agent has access to. In fact, if we just look at the lower bound $C^g$ in the Gaussian setting and consider each of the summand. Note that
> $T\_i^\top W\_i^g T\_i = \frac{\Sigma - \Pi\_i^\top \begin{bmatrix} 0 & 0 \\\ 0 & \Gamma\_{i-} \end{bmatrix} \Pi\_i}{\\|\theta\_{i-}\\|\_{\Gamma\_{i-}}^2 + \sigma^2}$. If we have strictly more coordinates observed for the $i$-th agent, then the Schur complement $\Pi\_i^\top \begin{bmatrix} 0 & 0 \\\ 0 & \Gamma\_{i-} \end{bmatrix} \Pi\_i$ will be a smaller matrix and also $\\|\theta_{i-}\\|\_{\Gamma_{i-}}^2$ will be a smaller quantity, resulting in an overall larger $T\_i^\top W\_i^g T\_i $ and thus smaller uncertainty $C^g$ and therefore also smaller test error. We will add this discussion in the camera ready version of the paper.
>
> **Response to o(n) bandwidth**: Thank you for bringing this up. You raise a good point about our storytelling with your observation that the local imputation baseline also has o(n) communication cost. While hopefully we were convincing in the introduction of why o(n) bandwidth constraints are important in real settings like satellites and seismic sensors, we agree that we did not adequately justify why a communication reduction from $d^2$ to $d\_i^2$ is important. We present this justification here now, and we will add it to the camera ready version. One property of a good collaborative algorithm is that agents which are a part of the collective should be incentivized to welcome new agents. In Collab, adding new agents to the collective never increases the communication cost to any of the existing agents a part of the collective. On the other hand, in the local imputation baseline algorithm, new agents with data collected from new/different sensors increase the communication cost of other agents in the collective (i.e., because $d$ becomes larger). This implicitly incentivizes homogeneity of sensors within the collective, which is antithetical to the idea of leveraging diverse data to make better predictions.
>
> **Response to the low-sample imputation underperformance**: This is a good observation. We believe the phenomenon boils down to numerical instability also known as the double descent for linear regression when the number of samples $n$ and underlying dimension $d$ is comparable. We point out that our theory is asymptotic and thus the statistical optimality holds for $n \\gg d$, and therefore not predictive in the low-sample regime. Though not within the scope of our current theoretical setup, it is nontheless an interesting future direction to investigate what is the optimal procedure when $n$ is comparable to $d$ (for instance, whether adding regularization would help). We will add this to the discussion section of our work.

---

> > ### Comment · Reviewer_Mws6 · 2023-08-16
> > **re: Rebuttal**
> >
> > Thank you for the detailed rebuttal. I will wait for the discussion period before forming my final score.
> >
> > That said, I think the paper could certainly improve in terms of presentation. The authors could do a better job in terms of convincing the reader why this problem is challenging (for instance, a discussion of why the 'first intuitive/natural solution' would not work, and why a better algorithm is necessary). They should also convey the proof intuitions and techniques better in the main text. This will help reviewers (especially non-experts like me) better appreciate the contributions.

---

### Official Review · Reviewer_jiy8 · 2023-07-23

**Soundness:** 4 excellent
**Presentation:** 3 good
**Contribution:** 4 excellent
**Rating:** 7
**Confidence:** 4

**Summary:**

This paper studied the problem of collaboratively learning least squares estimates with multiple agents, each of which only observes a different subset of the features. The authors proposed a distributed, semi-supervised algorithm called Collab consisting of three steps: 1) local training, 2) aggregation, and 3) distribution. The authors showed that the proposed Collab algorithm is nearly asymptotically local minimax optimal. The authors conducted experiments to verify their algorithm on real and synthetic data.

**Strengths:**

1. This paper provides deep theoretical insights into their proposed Collab algorithms. The results in Theorems 4.1 and 4.2 that the performance of Collab is no worse than local imputation with collaboration and global imputation are novel and surprising.

2. The asymptotic local minimax lower bound is interesting and could be of independent interest.

**Weaknesses:**

Although the results in this paper are interesting, as the authors themselves admitted, they are only limited to the linear model and Gaussian features. But the authors did provide some interesting discussions in Section 7 on future directions.

**Questions:**

1. In Step 2 of the main loop in the Collab algorithm, what happens if the estimate $\hat{\Sigma}_i$ is inaccurate? Could the authors analyze the impact of such errors?

2. Although the paper is well written in general, there are some minor typos. For example, in Line 162, it appears that the global estimator should be $\hat{\theta}^{clb}$.

---

> ### Author Rebuttal · Authors · 2023-08-06
>
> Thank you for your review.
>
> **Response to weaknesses**: Our theory is indeed limited to Gaussian features. We did experiments on non-Gaussian data in our Folktables experiment. Though it's only a preliminary experiment, we hope that it shows our method does not overfit to the Gaussian data setting.
>
> **Response to Q1**: noisy estimate of covariance: This is an interesting point.  We first want to point out a potential confusion that could be caused by a typo in the definition of $\hat{W}_i^g$ (line 155). The numerator is supposed to be the sample sub-covariance $\hat{\Sigma}\_{i+} = X\_{i+}^\top X\_{i+}/n$ instead of the exact sub-covariance matrix $\Sigma\_{i+}$. In fact, we use $\hat{\Sigma}\_{i+}$ in Algorithm 1 and in the proof of Corollary 3.2 in the submitted supplementary materials. It is not clear what is the optimal procedure if we do not have a consistent estimate for the population covariance, which essentially boils down to the harder problem of distributional shift. This could be a future direction and we will include it in our discussion.
>
> **Response to Q2: typos**:  Thanks for pointing this out. We will fix the typo you mentioned, the typo in the definition of $\hat{W}\_i^g$ (line 155), and other typos in the camera ready version of the paper.

---

> ### Comment · Area_Chair_psaR · 2023-08-16
> **Please acknowledge rebuttal**
>
> Dear reviewer,
>
> The authors have posted a rebuttal. Please acknowledge that you have read it and indicate whether they have adequately addressed your concerns/comments. The author-reviewer discussion phase ends on Aug 21 so please engage with the authors before that if you need any more clarifications.
> Thanks,
> AC

---

### Author Rebuttal · Authors · 2023-08-09

We want to clear up some possible confusion due to a typo we made. Our method Collab only needs the sample covariance $\hat{\Sigma}\_{i+} = X\_{i+}^\top X\_{i+}/n$ --- **not** the population covariance $\Sigma\_{i+}$ --- for our results to hold (see Algorithm 1 for the correct pseudocode). In other words, Collab does not need to know additional population information relative to the baselines. This potential confusion is likely caused by a typo in the definition of $\hat{W}\_i^g$ (line 155). The numerator is supposed to be the sample sub-covariance $\hat{\Sigma}\_{i+}$ instead of the exact sub-covariance matrix $\Sigma\_{i+}$. In fact, we use $\hat{\Sigma}\_{i+}$ in Algorithm 1 and in the proof of Corollary 3.2 in the submitted supplementary materials.

---

### Decision · Program_Chairs · 2023-09-21

**Decision:**

Accept (poster)

**Comment:**

Reviewers had mixed scores on this paper.  The paper has some merits: viz. the minimax optimality, but also suffers from the limitations reviewers have pointed out (viz. applying to a limited Gaussian covariates setting). However, overall it was felt that the merits outweigh the negatives, so the paper is accepted to NeurIPS. The authors are advised to carefully revise their paper with the feedback provided by the reviewers for the final version.